# Hyperosmotic stress memory in Arabidopsis is mediated by distinct epigenetically labile sites in the genome and is restricted in the male germline by DNA glycosylase activity

Anjar Wibowo[1†], Claude Becker[2†], Gianpiero Marconi[1,3], Julius Durr[1], Jonathan Price[1], Jorg Hagmann[2], Ranjith Papareddy[1], Hadi Putra[1], Jorge Kageyama[2], Jorg Becker[4], Detlef Weigel[2], Jose Gutierrez-Marcos[1]*

[1]School of Life Sciences, University of Warwick, Coventry, United Kingdom; [2]Department of Molecular Biology, Max Planck Institute for Developmental Biology, Tübingen, Germany; [3]Department of Agricultural, Food and Environmental Science, University of Perugia, Perugia, Italy; [4]Instituto Gulbenkian de Ciencia, Oeiras, Portugal

*For correspondence: j.f.gutierrez-marcos@warwick.ac.uk

†These authors contributed equally to this work

**Abstract** Inducible epigenetic changes in eukaryotes are believed to enable rapid adaptation to environmental fluctuations. We have found distinct regions of the Arabidopsis genome that are susceptible to DNA (de)methylation in response to hyperosmotic stress. The stress-induced epigenetic changes are associated with conditionally heritable adaptive phenotypic stress responses. However, these stress responses are primarily transmitted to the next generation through the female lineage due to widespread DNA glycosylase activity in the male germline, and extensively reset in the absence of stress. Using the *CNI1/ATL31* locus as an example, we demonstrate that epigenetically targeted sequences function as distantly-acting control elements of antisense long non-coding RNAs, which in turn regulate targeted gene expression in response to stress. Collectively, our findings reveal that plants use a highly dynamic maternal 'short-term stress memory' with which to respond to adverse external conditions. This transient memory relies on the DNA methylation machinery and associated transcriptional changes to extend the phenotypic plasticity accessible to the immediate offspring.

DOI: https://doi.org/10.7554/eLife.13546.001

## Introduction

While genetic variation is the primary source of long-term adaptation and evolution, numerous studies have pointed to induced epigenetic changes to facilitate rapid adaptation to short-term environmental fluctuations (*Franks and Hoffmann, 2012*). Because plants are sessile organisms, it has been suggested that they can efficiently integrate environmental signals into a 'stress memory' that is transmitted to the immediate progeny. This newly acquired information could allow populations to respond efficiently to repeated exposure to the same stress, a phenomenon known as 'priming' or 'acclimation' (*Crisp et al., 2016*; *Conrath et al., 2006*; *Sani et al., 2013*; *Slaughter et al., 2012*). It has recently been proposed that epigenetic marks could be induced depending on the consistency of the cues that individuals perceive directly from the environment (*Uller et al., 2015*). Yet it remains unclear how often this actually occurs in nature, and whether such adaptive responses can be

**eLife digest** Most plants spend their entire lives in one fixed spot and so must be able to quickly adapt to any changes in their surroundings. For example, high levels of salt in the soil – which can be toxic to cells – triggers stress responses in plants that help them to mitigate any damage. Once the stress has passed, plants are able to retain a memory of it, which allows them to respond more quickly if they face the same stress in future. Furthermore, plants may pass on this 'stress memory' to their offspring.

It is thought that stress memory is programmed by chemical modifications to DNA known as epigenetic marks. These marks do not alter the genetic information that is encoded by the DNA itself, but they can change the activity of particular genes. Environmental stress leads to changes in the epigenetic marks found on many plant genes, which can be directly passed on from the parent plant to its offspring. However, it was not clear whether the epigenetic marks that programme stress memory can be passed on in this way.

Wibowo, Becker et al. investigated how a model plant called *Arabidopsis thaliana* is able to remember periods of salt stress. The experiments show that high levels of salt can trigger changes in the patterns of epigenetic marks associated with particular regions of DNA. This memory is reinforced by repetitive exposure to similar salt stress and can be passed onto offspring, primarily through the maternal line. However, this stress memory is not fixed in future generations as the epigenetic marks can be reset to their original patterns if plants find themselves growing and reproducing under non-stress conditions.

In sum, the findings of Wibowo, Becker et al. show that epigenetic marks allow plants to inherit stress memory on a temporary basis while the stress is present, but to gradually lose the memory if the stress does not return. Future studies will focus on finding out if stress memory in crop plants works in the same way.

DOI: https://doi.org/10.7554/eLife.13546.002

transmitted over multiple non-stressed generations, a phenomenon termed "transgenerational stress memory" (*Hauser et al., 2011*; *Paszkowski and Grossniklaus, 2011*).

Stress memory in plants is believed to be mostly epigenetic in nature, because priming responses have been associated with changes in chromatin and DNA methylation (*Hauser et al., 2011*; *Ito et al., 2011*). Genome-wide studies in plants have shown that environmental stress dynamically modifies the chromatin landscape, creating novel patterns of gene expression, and thereby affecting short-term adaptation to stress (*Sani et al., 2013*). Moreover, heritable traits resulting from environmental stress have been associated with DNA methylation changes in promoter regions (*Bilichak et al., 2012*; *Le et al., 2014*), gene-coding regions (*Bilichak et al., 2012*; *Jiang et al., 2014*), transgenes (*Lang-Mladek et al., 2010*; *Molinier et al., 2006*) and transposable elements (TEs) (*Boyko et al., 2010*; *Le et al., 2014*; *Secco et al., 2015*).

Although TEs in plants are often mutagenic, they are nonetheless deemed to be potentially beneficial for regulating gene expression in response to a wide range of biotic and abiotic stresses. For instance, pathogen attack in Arabidopsis can cause changes in DNA methylation primarily at TEs and repeats located proximal to genes associated with transcriptional defense responses (*Dowen et al., 2012*), while temperature stress can trigger specific TE activation that, when inserted near genes, can confer stress-mediated transcriptional responses (*Cavrak et al., 2014*; *Naito et al., 2009*; *Yasuda et al., 2013*). Similarly in rice, phosphate starvation alters methylation at TEs near environmentally induced genes (*Secco et al., 2015*). These findings, which are reminiscent of the domestication of viral DNA for human immunity (*Chuong et al., 2016*), support the view that TEs play pivotal roles in environmental stress-sensing (*Kalendar et al., 2000*). The precise mechanism by which this occurs remains elusive. It is possible that the specific repetitive sequences present in transposons can generate and/or be recognised by stress-induced small non-coding RNAs that direct *de novo* DNA methylation (*Law and Jacobsen, 2010*). Alternatively, DNA methylation could be affected by stress independently of small RNAs through the RDR6-RdDM pathway (*Nuthikattu et al., 2013*) or be targeted and demethylated by DNA glycosylases (*Kim and Zilberman, 2014*; *Zhang and Zhu, 2012*), thereby imparting dynamic transcriptional changes at

neighbouring genes. This is supported by observations of mutants defective for genes of these epigenetic pathways, and which have impaired biotic and abiotic stress responses (*Boyko et al., 2010*; *Ito et al., 2011*; *Le et al., 2014*; *Luna and Ton, 2012*).

While the somatic stability of environmentally-induced epigenetic changes is well documented, robust evidence for their sexual transmission to the next generation is rare in both animals and plants (*Heard and Martienssen, 2014*; *Paszkowski and Grossniklaus, 2011*). During the initial stages of sexual reproduction, an active epigenetic reprogramming takes place in plant gametes as a means of silencing transposons (*Gutierrez-Marcos and Dickinson, 2012*; *Kawashima and Berger, 2014*), but it is not known whether this process also affects the transmission of environmentally-induced alterations in DNA methylation.

Here we report that repeated hyperosmotic stress induces DNA methylation changes that primarily affect epigenetically labile regions of the Arabidopsis genome, i.e., regions that are susceptible to changes in methylation status. Some of these changes are transmitted to the offspring, where they affect the transcriptional regulation of a small group of genes associated with enhanced tolerance to environmental stress. In the absence of a renewed stress stimulus, the acquired epigenetic and phenotypic changes are gradually reset in subsequent generations. Further, newly acquired stress tolerance and associated *de novo* DNA methylation marks are preferentially transmitted through the female germline. Epigenetic inheritance relies on DNA methylation changes at sequences that function as distantly acting control elements of key stress-response regulators, including antisense long non-coding RNAs (lncRNAs). Collectively, our data provide a new mechanistic model for the establishment and transient inheritance of plant stress adaptation.

## Results

### Repeated hyperosmotic stress leads to transient phenotypic adaptation

To evaluate the extent to which stress-induced transgenerational adaptation occurs, we first exposed Arabidopsis plants (*Arabidopsis thaliana* accession Col-0) to two different hyperosmotic conditions for over five generations (P0 of G1-G5, *Figure 1A*) (see Materials and methods for details). Plants were subjected to stress only during the vegetative phase and were transferred to normal soil before most flowers formed; thereby reducing the possibility of parental stress exerting a direct effect on gametes (see Materials and methods for details). To uncouple parent and progeny environments, we grew seeds derived from G1-G5 treated and control plants for two additional generations (P1 and P2) without stress (*Figure 1A*). Germination and survival rates of the progeny were then assessed in three independent experiments for all generations under stress and control conditions.

Similar to non-treated plants, the progeny derived from the first generation of stressed plants (G1) did not display significant signs of adaptation. In contrast, the direct progeny (P1) of G2-G5 stressed plants displayed higher germination and survival rates, and more robust vegetative growth on high-salinity (150 mM) medium (*Figure 1B–C*, *Figure 1—figure supplement 1* and *Supplementary file 1*), suggesting that hyperosmotic priming requires repetitive exposure to stress. Notably, adaptation was already lost in the second-generation progeny (P2) of G2-G5 stressed plants. Thus, two generations of a stress-free environment were sufficient to revert the stress-induced changes (*Figure 1C* and *Figure 1—figure supplement 1*). Our data thus suggest that recurrent hyperosmotic stress in plants induces intergenerational adaptation, but that this response does not persist in the absence of stress.

### Transient adaptation to environmental stress is regulated epigenetically

Several studies have suggested that environmental stress induces genome-wide epigenetic changes that can be transmitted to the offspring (*Boyko et al., 2010*; *Jiang et al., 2014*; *Luna et al., 2012*; *Rasmann et al., 2012*; *Slaughter et al., 2012*). For this to occur, such changes must escape the epigenetic reprogramming that takes place during sexual reproduction (*Calarco et al., 2012*; *Ibarra et al., 2012*). Therefore, to confirm that the adaptive responses seen in the offspring of stressed parents were due to newly acquired genome-wide epigenetic changes, we grew well-characterized epigenetic mutants that are defective in RNA-directed DNA methylation (RdDM) or in the active removal of DNA methylation (*Law and Jacobsen, 2010*; *Zhang and Zhu, 2012*), and exposed

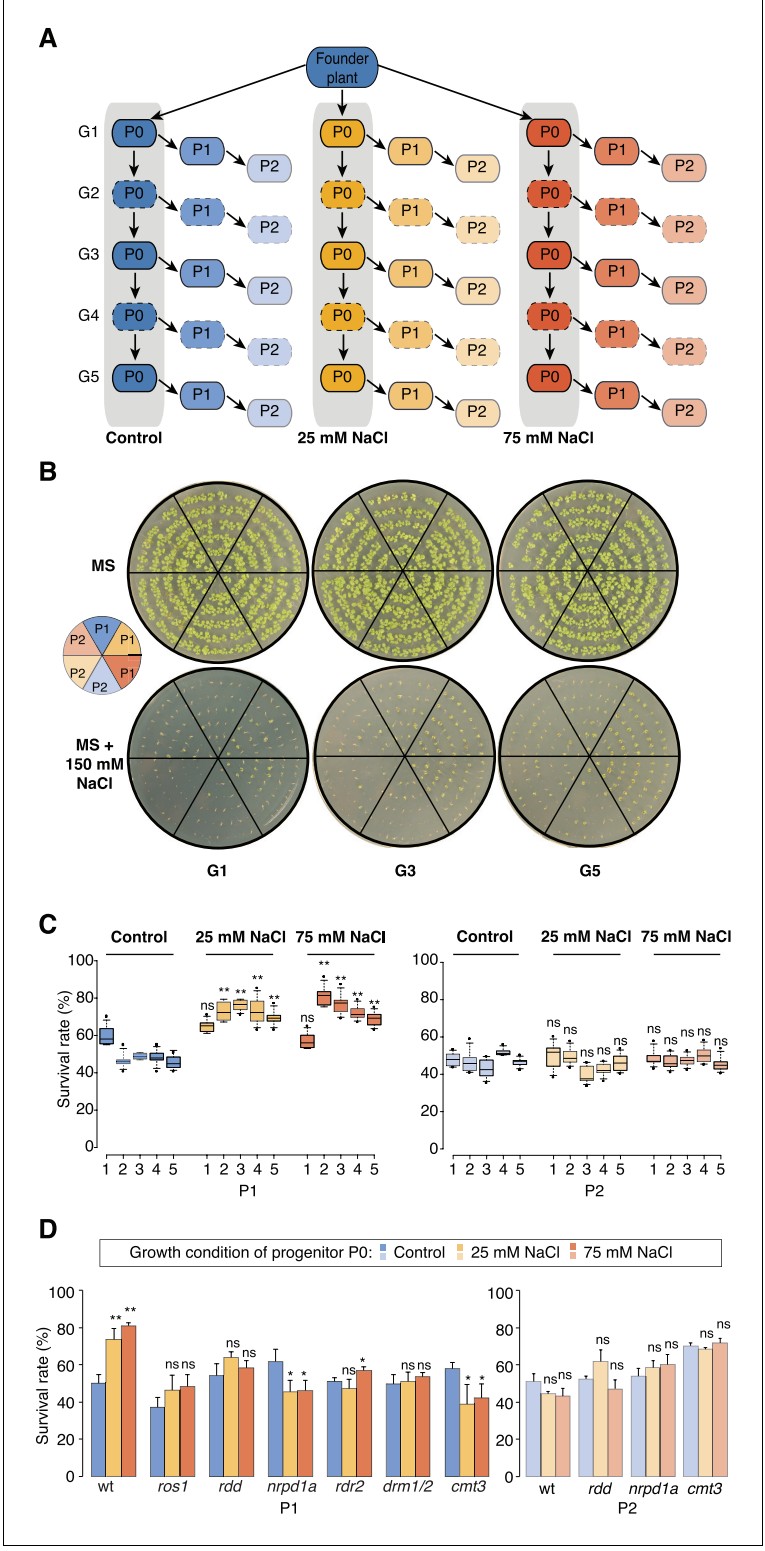

**Figure 1.** Improved salt resistance of progeny from parents exposed to hyperosmotic stress. (**A**) Diagram of the experimental design. Plants were grown on control or hyperosmotic medium (25 mM and 75 mM NaCl) for five consecutive generations. From each generation, progeny in P1 and P2 were grown in the absence of stress. Pools of 10 plants from generations 1, 3 and 5, and of their respective P1 and P2 progeny were used for bisulfite sequencing (solid-lined small boxes). (**B**) Salt tolerance assay of P1 and P2 progeny of control and G1, G3 and G5 salt-treated plants. Seeds were germinated on MS with or without NaCl; images are 2 weeks after sowing. Color

*Figure 1 continued on next page*

*Figure 1 continued*
code as in (**A**). (**C**) Survival of P1 and P2 seedlings grown on medium with 150 mM NaCl. At least 300 seedlings tested per triplicate in two independent experiments. Asterisks indicate a significant difference between the control group of the same generation (unpaired Student's *t*-test; * p<0.05, ** p<0.01, ns p>0.05). Horizontal bar corresponds to median, whiskers indicate entire 95th percentile. (**D**) Survival of wild-type (wt) and RdDM and DNA methylation mutant P1 and P2 seedlings on medium with 150 mM NaCl (unpaired Student's *t*-test; * p<0.05, ns p>0.05). Error bars indicate standard deviation.
DOI: https://doi.org/10.7554/eLife.13546.003
The following figure supplement is available for figure 1:

**Figure supplement 1.** High-salinity tolerance assays.
DOI: https://doi.org/10.7554/eLife.13546.004

them to hyperosmotic stress for two successive generations. We assessed the progeny for enhanced tolerance to high salinity in three independent experiments (*Figure 1D*). In contrast to progeny of stressed wild-type plants, immediate progeny of stressed *nrpd1a* (*Herr et al., 2005*) *cmt3* (*Chan et al., 2006*) and *ros1/dml2/dml3 (rdd)* (*Penterman et al., 2007*) plants did not show enhanced survival under hyperosmotic stress conditions (*Supplementary file 1*). These data imply that transgenerational adaptation to hyperosmotic stress relies in part on the DNA methylation machinery, although these phenotypic data do not reveal how extensive the epigenetic changes are.

## Hyperosmotic stress leads to distinct DNA methylation changes

To determine the primary genomic targets susceptible to epigenetic changes induced by hyperosmosis, we performed whole-genome bisulfite sequencing (*Supplementary file 2*). To ensure statistically robust results, we excluded inter-individual epigenetic variation that can arise over the course of several generations (*Becker et al., 2011*; *Schmitz et al., 2011*) by collecting duplicate samples of leaf tissue from 10 plants each from the G1, G3 and G5 generations for each treatment (control, 25 mM-, and 75 mM-NaCl) (*Figure 1A*). We sought to compare DNA methylation patterns for the different treatments in non-stressed P1 and P2 progeny derived from control or salt-stressed P0 parents (*Figure 1A*). Individual cytosines with a significantly altered methylation frequency, termed differentially methylated positions (DMPs), were first identified by pairwise comparisons between two samples. Because the three stressed generations had been grown and treated at different time points, we only compared samples belonging to the same treatment group, thus excluding methylation changes that were due to stochastic fluctuations in growth conditions. Single-site polymorphisms between any two samples were rare, with on average 6,866 DMPs (40% CG, 15% CHG and 45% CHH) detected per generation in all pairwise comparisons (*Supplementary file 3*). Principal component analysis (PCA) and complete linkage clustering of methylation frequencies grouped all stress-treated samples (P0), separating them from control, P1, and P2 samples (*Figure 2—figure supplement 1A–B*). This indicated that hyperosmotic stress had a small but noticeable effect on single methylated cytosines, and that this effect was largely transient. Overall we observed three times more methylation gains than losses in salt-treated P0 plants compared to the control (*Figure 2—figure supplement 1C*). Because we found considerably fewer DMPs than recently reported for multi-generational hyperosmotic stress treatments (*Jiang et al., 2014*), we re-analysed the published data. We found that only a small fraction of the DNA methylation changes reported by *Jiang et al. (2014)* were consistently induced by hyperosmotic stress (*Figure 2—figure supplement 1D*).

The properties of DMPs are distinct from those of differentially methylated regions (DMRs, i.e., contiguous stretches of methylation change), as DMPs mostly occur at sparsely distributed CG sites within gene bodies, whereas DMRs tend to occur in densely methylated areas of mixed methylation context (*Hagmann et al., 2015*). To identify stress-induced DMRs, we used a statistically robust Hidden Markov Model-based algorithm that supports the confident detection of differential methylation also in CHG and CHH contexts (*Hagmann et al., 2015*). We identified on average 24,700 methylated regions (MRs) per sample with a median length of 272 bp (mean: 856 bp). To identify generation-specific and treatment-dependent DMRs, we considered samples of the same generation (G1, G3 or G5, *Figure 1A*) and treatment regime (control or salt-treated) as replicates (*Supplementary file 4*). For all three generations, DMRs mapped mainly to TEs and intergenic regions, and were three- to seven-fold over-represented in 2-kb regions upstream of transcription

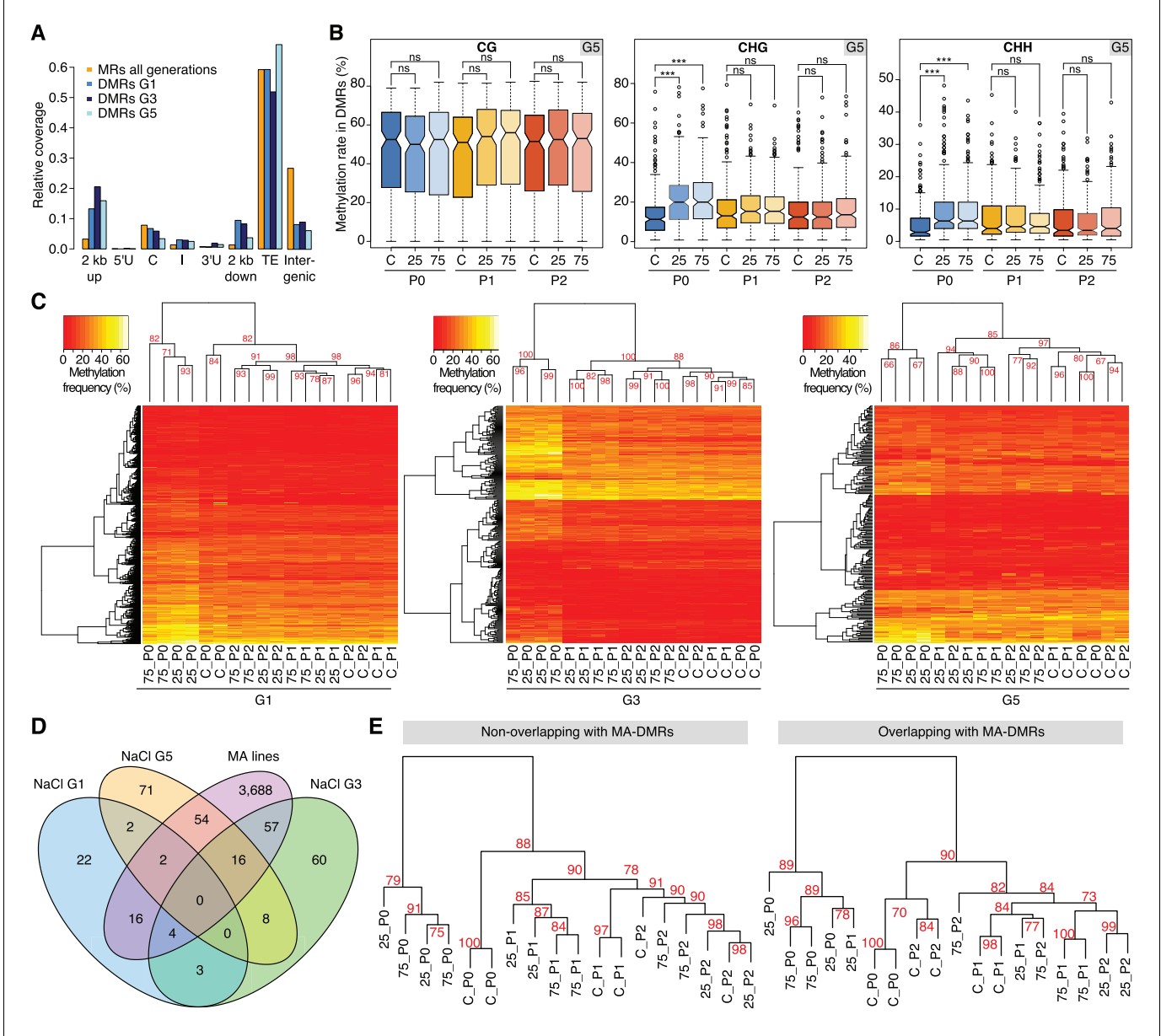

**Figure 2.** Hyperosmotic stress-induced differentially methylated regions (DMRs) in the absence of stress stimulus. (**A**) Annotation of cytosines in MRs and DMRs between P0 control and P0 hyperosmotic treated samples in different generations (see *Figure 1A*). (**B**) Methylation frequencies by sequence context in DMRs identified between control (P0 of G5), stress-treated (P0 of G5), and the derived P1 and P2 plants (unpaired two-tailed Student's *t*-test; ***p<0.001, ns p>0.05). Horizontal bar corresponds to median, whiskers indicate entire 95th percentile. (**C**) Complete linkage clustering of samples from different generations based on DMR methylation frequencies. Methylation frequency of cytosines contained in each DMR were averaged, and only DMRs covered in all samples were considered. Numbers in red indicate approximately unbiased (AU) *p*-values (x100), calculated with *pvclust*. (**D**) Overlap (including 500 bp flanking windows) of DMRs between P0 control and stress-treated samples from G1, G3 and G5. Overlap with DMRs from a previous analysis of mutation accumulation (MA) lines (*Hagmann et al., 2015*) is also shown. (**E**) Clustering of DMRs between P0 control and stress-treated samples in G5 according to overlap with MA-DMRs. C, control, 25, 25 mM NaCl and 75, 75 mM NaCl.

DOI: https://doi.org/10.7554/eLife.13546.005

The following figure supplements are available for figure 2:

**Figure supplement 1.** DNA methylation variation after multigenerational hyperosmotic stress.
DOI: https://doi.org/10.7554/eLife.13546.006

**Figure supplement 2.** Effect of hyperosmotic-stress on global methylation.
DOI: https://doi.org/10.7554/eLife.13546.007

**Figure supplement 3.** Hyperosmotic-stress induced methylation changes in DMRs.

*Figure 2 continued on next page*

*Figure 2 continued*

DOI: https://doi.org/10.7554/eLife.13546.008

**Figure supplement 4.** Overlap of HS-DMRs with MA-DMRs.

DOI: https://doi.org/10.7554/eLife.13546.009

**Figure supplement 5.** Gene Ontology analysis of genes associated to HS-DMRs.

DOI: https://doi.org/10.7554/eLife.13546.010

start sites compared with overall methylation in MRs (*Figure 2A*). No significant differences were found in the average methylation frequencies at MRs in control and stressed samples (*Figure 2—figure supplement 2*), or for CG methylation frequencies at DMRs in control and stress-treated samples ($P > 0.05$ in all generations, unpaired two-tailed Student's *t*-test) (*Figure 2B*; *Figure 2—figure supplement 3*). However, methylation in CHG and CHH contexts at DMRs was significantly altered in stress-treated P0 versus control P0 samples ($P < 0.01$ in all generations, unpaired two-tailed Student's *t*-test) (*Figure 2B*; *Figure 2—figure supplement 3*).

Complete linkage clustering of DMRs grouped salt-treated P0 samples in all three generations (*Figure 2C*), similar to the clustering and Principal Component Analysis (PCA) on single polymorphic sites (*Figure 2—figure supplement 1A–B*). For G1, the cluster comprising the non-stressed samples did not have any clear substructure. By contrast, for G3 and G5, the P1-descendants of salt-treated P0 plants formed a clear sub-group distinct from the control and P2 plants (*Figure 2C*). These data concur with the adaptive responses we observed specifically in the P1, but not in the P2 progeny of G3 and G5 salt-treated plants (*Figure 1C*; *Figure 1—figure supplement 1A–B*). P2-descendants showed methylation patterns similar to control plants, which correlated with the observed lack of high salinity tolerance. We also analysed published DNA methylation data derived from individual plants subjected to hyperosmotic stress (*Jiang et al., 2014*), focussing on DMRs, and considering individual samples as replicates. We confirmed our finding that the methylation changes in CHG and CHH correlated well with stress treatment, whereas changes in CG methylation did not, indicating that CG methylation patterns occur stochastically in treated and non-treated samples (*Figure 2—figure supplement 1E*). Thus, hyperosmotic stress directs DNA methylation changes primarily at non-CG sites located in intergenic TE-related sequences, and these epigenetic modifications are associated with an acquired transient adaptation to stress.

## Identification of epigenetically labile regions sensitive to abiotic stress

Although a large fraction of DMRs appeared to arise as a consequence of salt treatment, only a few recurred in G3 and G5, or overlapped between all three generations (*Figure 2D*). To determine the significance of these hyperosmotic-stress DMRs (HS-DMRs), we asked whether they were also responsive to other abiotic stresses, such as cold treatment. Comparing our data with a small set of DMRs in cold-stressed Arabidopsis seedlings (*Seymour et al., 2014*), we did not detect any overlap between the two datasets. By contrast, 49% of HS-DMRs in stressed plants (P0) overlapped with or were in close proximity (<500 bp) to DMRs that had been reported to arise spontaneously in mutation accumulation (MA) lines that had been grown over 30 generations under controlled conditions (*Figure 2D*) (*Becker et al., 2011*; *Hagmann et al., 2015*). These spontaneous MA-DMRs are often found in more than one individual, pointing to specific regions of the genome being particularly susceptible to epigenetic reprogramming. We compared pools of ten plants to identify HS-DMRs, while individual plants had been compared to define MA-DMRs. That there is nevertheless substantial overlap indicates that stress-triggered epigenetic reprogramming of specific genomic regions is not entirely random.

To determine whether the DMRs found in both the hyperosmotic stressed plants and MA lines differed from those others found in only one of the two populations, we performed complete linkage clustering on both sets. Overlapping and non-overlapping DMRs behaved similarly for G3 and G5 (but not G1), with stressed samples clustering in one group and control and untreated progeny in another. This indicated that overlapping and non-overlapping DMRs carried a similar hyperosmotic stress signature (*Figure 2E*; *Figure 2—figure supplement 4*), but the two DMR classes differed in their association with adjacent annotated genes (*Figure 2—figure supplement 5*). In particular, HS-DMRs that did not overlap with spontaneous MA-DMRs were enriched near genes with functions related to metabolic responses and ion transport (*Figure 2—figure supplement 5*).

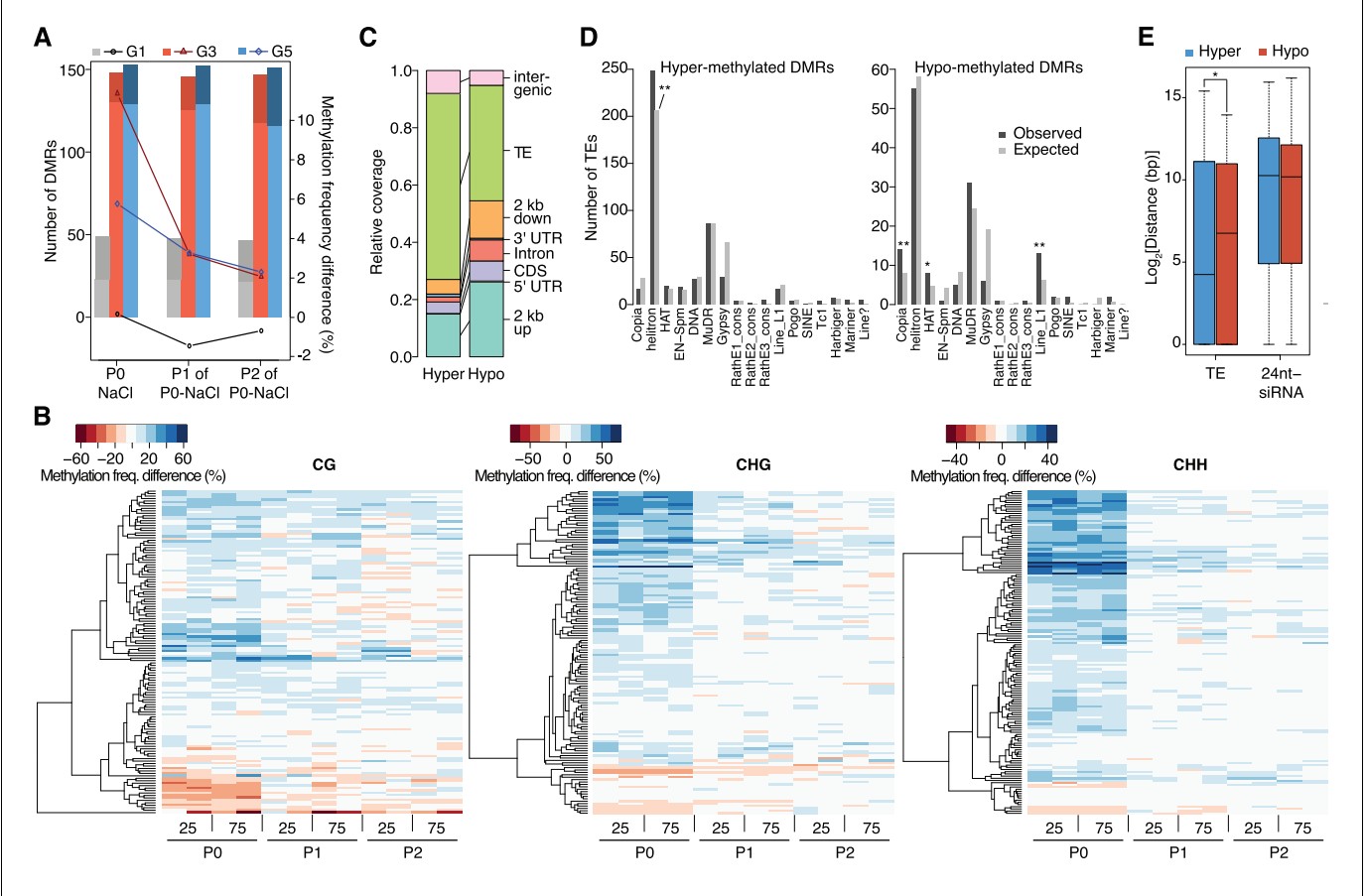

**Figure 3.** Dynamics of methylation frequency changes in DMRs. (**A**) DMRs that are hypo- (darker colours) or hyper-methylated (lighter colours) in stress-treated P0 and their P1 and P2 progeny compared to the average of all control samples. Methylation states were determined by subtracting the methylation frequency of a DMR in the respective sample from the combined controls (P0, P1 and P2) of that generation (G1, G3, or G5); positive differences were considered as hyper-, negative differences as hypo-methylation events. Line plots indicate the absolute net methylation frequency change (in% ) across all DMRs. (**B**) One-directional clustering of DMRs in G3 by methylation frequency difference, separated by sequence context. Differences for each DMR were calculated by subtracting the methylation frequency of the DMR in a sample from the average in the combined P0, P1 and P2 control samples (see also *Figure 1a*). Blue colour indicates hyper-, red colour indicates hypomethylation. 25, 25 mM NaCl; 75, 75 mM NaCl; C, control. (**C**) Annotation of DMRs. (**D**) Classes of TEs next to DMRs. (**E**) Distance of DMRs to the nearest transposable element (TE) or 24 nt-siRNA locus (*Fahlgren et al., 2010*) (unpaired Student's *t*-test; *p<0.05). Horizontal bar corresponds to median, whiskers indicate entire 95th percentile; outliers are not shown.

DOI: https://doi.org/10.7554/eLife.13546.011

The following figure supplements are available for figure 3:

**Figure supplement 1.** Methylation dynamics of HS-DMRs across three generations.

DOI: https://doi.org/10.7554/eLife.13546.012

**Figure supplement 2.** Methylation at hyperosmosis-induced DMRs in *drm1drm2* double mutants.

DOI: https://doi.org/10.7554/eLife.13546.013

These data thus suggest that exposure to hyperosmotic stress targets discrete, epigenetically labile regions of the genome.

## Stress induces transient hypermethylation at selected transposon-sequences

In contrast to previous reports suggesting wholesale DNA methylation changes induced by environmental stress (*Boyko et al., 2010*; *Dowen et al., 2012*; *Jiang et al., 2014*), we noticed that methylation frequencies within HS-DMRs differed between generations. Most HS-DMRs (81%) in G3 and G5 were hypermethylated in the hyperosmotic-treated P0 samples, whereas G1 plants had similar number of hyper- and hypo-methylated DMRs (*Figure 3A*). Further, in progenies of stress-treated plants,

**Table 1.** Association between differential DNA methylation induced by hyperosmotic stress and histone modifications (*Sani et al., 2013*).

| Chromatin mark | Hypo HS-DMRs | Hyper HS-DMRs | MRs |
|---|---|---|---|
| All | 70 | 280 | 72,074 |
| H3K4me2 (650/98) | 0/0 | 3/0 | 508/155 |
| H3K4me3 (1454/46) | 4/0 | 8/0 | 1065/40 |
| H3K9me3 (276/254) | 0/0 | 0/0 | 484/920 |
| H3K27me3 (1213/6520) | 3/15 | 7/99 | 1318/7774 |

Intersections between high salt-induced DMRs (HS-DMRs), MRs and different chromatin marks (*Sani et al., 2013*). The two numbers behind each mark indicate significantly increased or decreased regions after hyperosmotic stress (*Sani et al., 2013*).

DOI: https://doi.org/10.7554/eLife.13546.014

the CHG and CHH methylation changes were gradually lost and reverted to the control states, as seen in the P1 and P2 progeny grown in the absence of stress (*Figure 3A–B*; *Figure 3—figure supplement 1*). Hyper- and hypo-methylated HS-DMRs mapped to different genomic regions, with methylation gain after hyperosmotic stress frequently found within or proximal to TEs, but methylation loss occurring more frequently near genes (*Figure 3C*). Hyper-methylated HS-DMRs were significantly enriched near *Helitrons* (unpaired Student's *t*-test; **p<0.01) (*Figure 3D*), a TE family known to be targeted by RdDM (*Nuthikattu et al., 2013*), and near genes involved in RNA-directed DNA polymerase and reverse transcription activities (*Figure 2—figure supplement 5*). In contrast, hypo-methylated DMRs were found proximal to *Copia, HAT,* and *Line L1* TEs (*Figure 3D*). These transposon families are targeted by DNA demethylases and are associated with gene expression response to environmental stress (*Le et al., 2014*). By contract, an increase in non-CG hyper-methylation is indicative of RdDM activity (*Law and Jacobsen, 2010*). To test whether RdDM was responsible for some of the HS-DMRs, we exposed *drm1/drm2* double mutants to hyperosmotic stress over two consecutive generations. In contrast to wild-type, *drm1/drm2* plants did not show hyper-methylation in non-CG contexts (*Figure 3—figure supplement 2*). However, when we analysed public datasets for known siRNA loci (*Fahlgren et al., 2010*), we could not detect a correlation between methylation status and the presence of active siRNA production (*Figure 3E*).

Because hyperosmotic priming has been associated with discrete changes in the chromatin landscape (*Sani et al., 2013*), we assessed the relationship between induced changes in DNA methylation and chromatin marks. Although hyperosmotic stress and growth conditions used by *Sani et al., 2013* differed from ours, we found that HS-DMRs were enriched for hyperosmosis-altered tri-methylated lysine 27 in histone 3 (H3K27me3) compared to background MRs (63% vs. 13%) (*Table 1*). Notably, hypermethylated HS-DMRs (38%) were associated with decreased H3K27me3, revealing an antagonistic relationship between these two repressive epigenetic marks in discrete genome regions involved in hyperosmotic priming. Collectively, our data show that hyperosmotic stress induces transient DNA methylation and chromatin changes at intergenic elements derived from specific transposon families.

## Stress-directed adaptation shows biased sexual transmission

Because adaptive stress responses in plants have been proposed to be largely under maternal control (*Agrawal, 2001*; *Pecinka and Mittelsten Scheid, 2012*), we investigated the mode of inheritance of the enhanced tolerance to hyperosmotic stress by reciprocally crossing stressed and unstressed plants. We found that enhanced tolerance to hyperosmotic stress conditions was primarily conferred through the female germline (*Figure 4A*). As the epigenetic reprogramming of male gametes is mediated to a large extent by DEMETER (DME) DNA glycosylase activity (*Calarco et al., 2012*; *Ibarra et al., 2012*), we investigated whether hyperosmotic priming could be passed on through the male germline if DME-dependent reprogramming was disrupted. We stressed *dme-6* plants for two generations and tested the offspring for tolerance to hyperosmotic stress. The progeny of *dme-6* stressed plants were even more tolerant to hyperosmotic stress (Student's *t*-test, p<0.001) than that of stressed control plants (*Figure 4A*), thus implicating DME's involvement in resetting stress-directed methylation marks in the male germline. To confirm this hypothesis, we

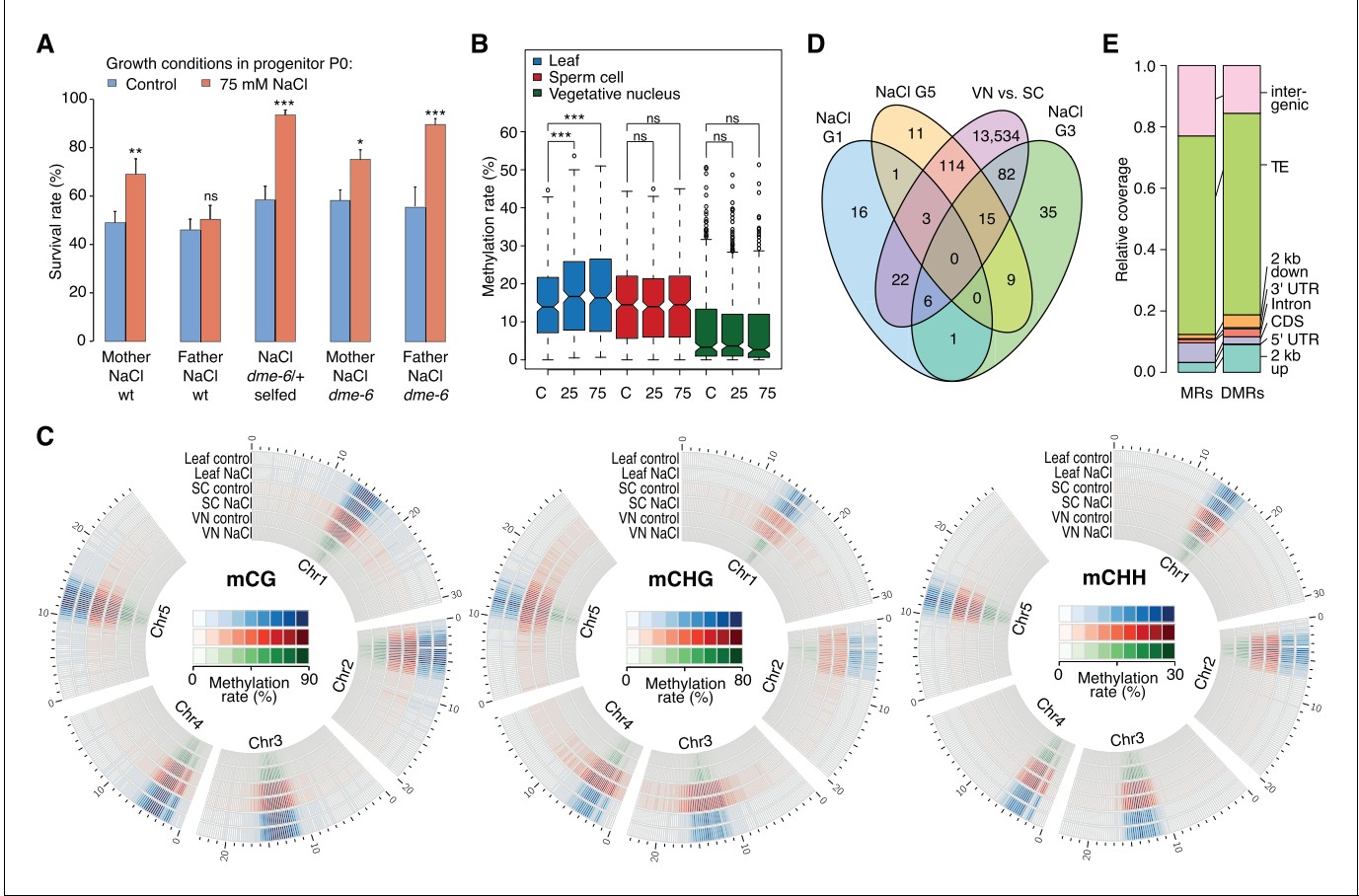

**Figure 4.** Parent-of-origin effects on stress-induced epimutations. (**A**) Survival of F₁ seedlings derived from reciprocal crosses between Col-0 wild type or *dme-6* mutants that had been exposed to hyperosmotic stress for two generations and untreated wild-type (wt), or progeny of *dme-6*/+ selfed plants (unpaired Student's *t*-test; *p<0.05, **p<0.01, ***p<0.001, ns p>0.05). (**B**) Absolute methylation frequency differences in DMRs in different tissues from control and stress-treated plants (unpaired, two-sided Student's *t*-test; ***p<0.001, ns p>0.05). C, control; 25, 25 mM NaCl; 75, 75 mM NaCl. (**C**) Genome-wide methylation levels in leaves and pollen derived from control and salt-stressed P0 plants (generation G1). Methylation frequency was calculated as the average methylation frequency of cytosines in a 250 kb window. Chr, chromosome. (**D**) Overlap of DMRs from the comparison of vegetative nuclei and sperm cells with DMRs identified in leaf tissue after salt treatment in G1, G3, G5. (**E**) Annotation of MRs and DMRs in vegetative nuclei and sperm cells.

DOI: https://doi.org/10.7554/eLife.13546.015

The following figure supplements are available for figure 4:

**Figure supplement 1.** Isolation of sperm cells and vegetative nuclei by fluorescent-activated-cell-sorting.

DOI: https://doi.org/10.7554/eLife.13546.016

**Figure supplement 2.** Methylation at hyperosmotic stress-induced DMRs in the *dme-6* mutants.

DOI: https://doi.org/10.7554/eLife.13546.017

performed reciprocal crosses between unstressed wild-type and stressed heterozygous *dme*-6 plants (*Figure 4A*). The differential responses observed confirmed that DME diminishes the paternal transmission of hyperosmotic priming responses.

To define the magnitude of the transmission of newly acquired epigenetic marks through the male germline, we compared the methylomes of sperm cell (SC) and vegetative nuclei (VN) isolated from control and hyperosmotic-stressed plants (*Figure 4—figure supplement 1* and *Supplementary file 2*). Unlike in somatic tissue, hyperosmotic stress induced very few methylation changes in SC and VN (*Figure 4B*) such that only three DMRs became hypomethylated upon hyperosmotic treatment. This strikingly contrasted with the13,776 DMRs that distinguished SCs from VNs (SV-DMRs) (*Figure 4C*). Most of the SV-DMRs predominantly localized next to TEs and were particularly enriched in adjacent regions of coding sequences (*Figure 4D*). Significantly, over three quarters

of the HS-DMRs (76%) overlapped with SV-DMRs (*Figure 4C*) and half of them with MA-DMRs (*Figure 2D*). Further, when compared with somatic tissues, CHG methylation in SC was elevated in pericentromeric regions, while methylation in VN was depleted in all sequence contexts and reduced to the central centromeric region (*Figure 4E*). Because siRNAs produced in the vegetative nuclei can silence transposons in sperm cells in a DME-dependent process (*Ibarra et al., 2012*; *Duan et al., 2016*), we assessed the methylation state of HS-DMRs in *dme* mutants. Most HS-DMRs were hyper-methylated in *dme*-6 sperm cells but hypo-methylated in *dme*-6 vegetative nuclei (*Figure 4—figure supplemental 2*). However, we observed similar methylation differences at non-DMR coordinates (*Figure 4—figure supplemental 2*), indicating that HS-DMRs are only a subset of loci that are under the control of DME.

In summary, DNA glycosylase activity in the male germline is pivotal for both the epigenetic silencing of transposons and for the resetting of epigenetic marks induced by environmental stress. As a consequence, the hyperosmotic priming effects are unequally transmitted through the male and female germlines.

## Stress-directed epimutations are associated with reprogramming of the salt-responsive transcriptome

It has been proposed that abiotic stress can lead to heritable epigenetic changes particularly affecting transgenic repeats (*Lang-Mladek et al., 2010*; *Molinier et al., 2006*) and TEs (*Bilichak et al., 2012*; *Ito et al., 2011*), and possibly the expression of neighbouring genes (*Dowen et al., 2012*; *Wang et al., 2013*). In agreement, HS-DMRs were frequently identified in transposon-related sequences and over-represented in regions proximal to protein-coding genes (*Figure 2A*), suggesting that salt-induced methylation changes might be linked to differential gene expression. We found 123 genes that were flanked by HS-DMRs (*Figure 2D*) of which one third (32%) have been previously shown to be responsive to osmotic stress (*Zeller et al., 2009*) (*Figure 5—figure supplement 1* and *Supplementary file 5*).

We identified one HS-DMR overlapping two TEs (*Figure 5A*) that was found upstream of *MYB DOMAIN PROTEIN 20* (*MYB20*), which encodes a transcription factor involved in abscisic acid (ABA) signalling and implicated in stress tolerance (*Cui et al., 2013*). This HS-DMR became hyper-methylated in P0 plants exposed to hyperosmotic stress, which was maintained in the P1 progeny of stressed plants, but then changed to levels similar to that seen in control plants in the P2 progeny. We did not detect *MYB20* expression changes in response to high hyperosmotic treatment in plants whose progenitors had not experienced hyperosmotic stress (*Figure 5B*). However, when P0 plants had experienced such stress, this gene was strongly downregulated in P1 and P2 progeny (*p*-value 0.006) (*Figure 5B*).

Another HS-DMR was located downstream of the *CARBON/NITROGEN INSENSITIVE 1* (*CNI1*) gene (*Figure 5A*), which encodes a membrane RING-type ubiquitin ligase implicated in metabolic sensing (*Sato et al., 2009*). The *CNI1* HS-DMR, which also overlapped a TE, had reduced DNA methylation in the P1 progeny of stressed plants, and remained hypomethylated in P2 progeny. Hyperosmotic stress strongly reduced *CNI1* expression in progeny of untreated plants, and to a lesser extent in progeny of plants exposed to hyperosmotic stress (*Figure 5C*). We also analysed four additional genes with adjacent HS-DMR and found similar modes of epigenetic regulation and inheritance (*Figure 5—figure supplement 2*).

We then examined whether the stress responsiveness of these genes were altered in epigenetic mutants that were immune to hyperosmotic stress adaptation (*Figure 1D*). Wild-type and mutant plants were grown in control or hyperosmotic stress conditions for two generations, and gene expression was analysed in P1 progeny exposed to stress. Independent of growth condition or priming, the expression of *MYB20, CNI1* and four other genes with associated HS-DMRs was altered in *rdd* demethylation mutants, and the transcriptional stress response in P1 progeny of stressed plants was impaired in RdDM mutants (*Figure 5B–C*; *Figure 5—figure supplement 2*). We then assessed the transcriptional response of genes encoding components of the DNA methylation and demethylation pathways and found that many of them were sensitive to hyperosmotic salt treatment (*Figure 5—figure supplement 3*). Moreover, analysis of epigenomic data for RdDM and demethylation mutants (*Stroud et al., 2013*) revealed that methylation at regions corresponding to HS-DMRs from our data set was severely affected in *met1* and *rdd* mutants (Fischer's Exact test p=0,005) (*Figure 5—figure supplement 4*).

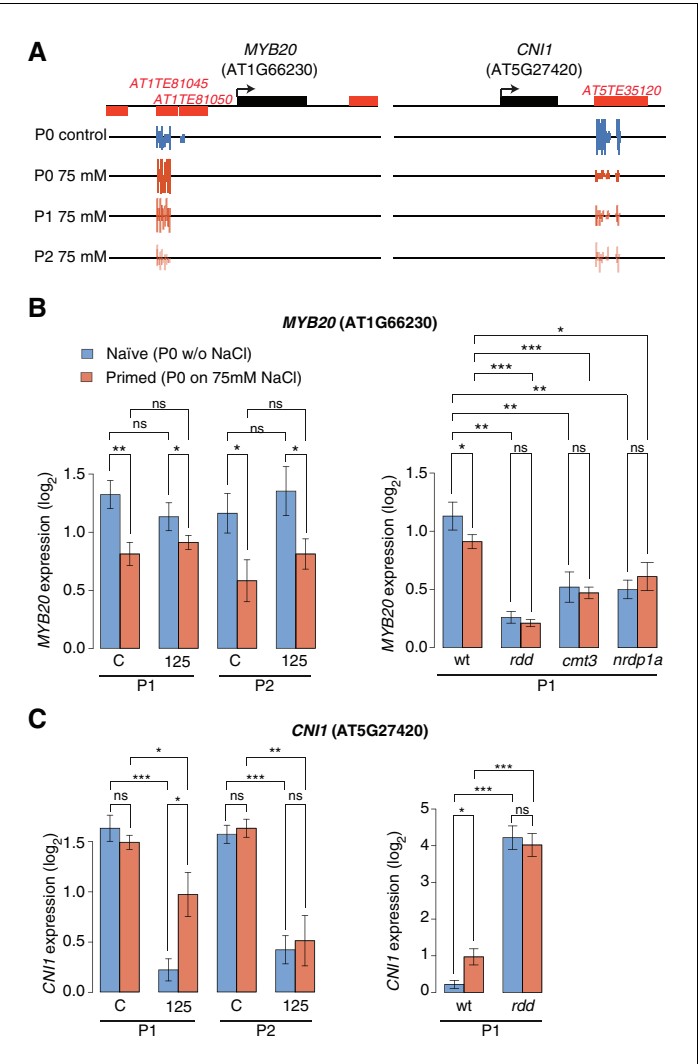

**Figure 5.** Expression of two genes adjacent to hyperosmotic stress-induced DMRs. (**A**) Methylation near *MYB20* and *CNI1*. Black boxes on top represent genes, red boxes TEs. Methylation on the top and bottom strands at individual cytosines is shown as vertical bars below. (**B–C**) *MYB20* and *CNI1* expression (arbitrary units) in the P1 and P2 progeny of control ('naïve') and salt-treated ('primed') wild-type and mutant plants. Leaves of 2-week-old plants grown on MS medium were analysed (unpaired Student's *t*-test; *p<0.05, **p<0.01, ns p>0.05). Error bars indicate standard deviation. C, control; 125, 125 mM NaCl.

DOI: https://doi.org/10.7554/eLife.13546.018

The following figure supplements are available for figure 5:

**Figure supplement 1.** Hyperosmotic stress response of genes next to HS-DMRs.
DOI: https://doi.org/10.7554/eLife.13546.019

**Figure supplement 2.** Expression of genes adjacent to HS-DMRs.
DOI: https://doi.org/10.7554/eLife.13546.020

**Figure supplement 3.** Expression of RdDM and demethylation pathway genes in response to hyperosmotic stress.
DOI: https://doi.org/10.7554/eLife.13546.021

**Figure supplement 4.** Methylation profiles of HS-DMRs in DNA methylation and demethylation Arabidopsis mutants.
DOI: https://doi.org/10.7554/eLife.13546.022

We focused our attention on the *CNI1* HS-DMR due to its unusual location downstream of the transcription unit. We hypothesised that this HS-DMR may act as a long-distance regulatory element. We therefore analysed the effects of two independent mutant alleles, in which T-DNAs were inserted between the *CNI1* transcription unit and the downstream HS-DMR, possibly impeding

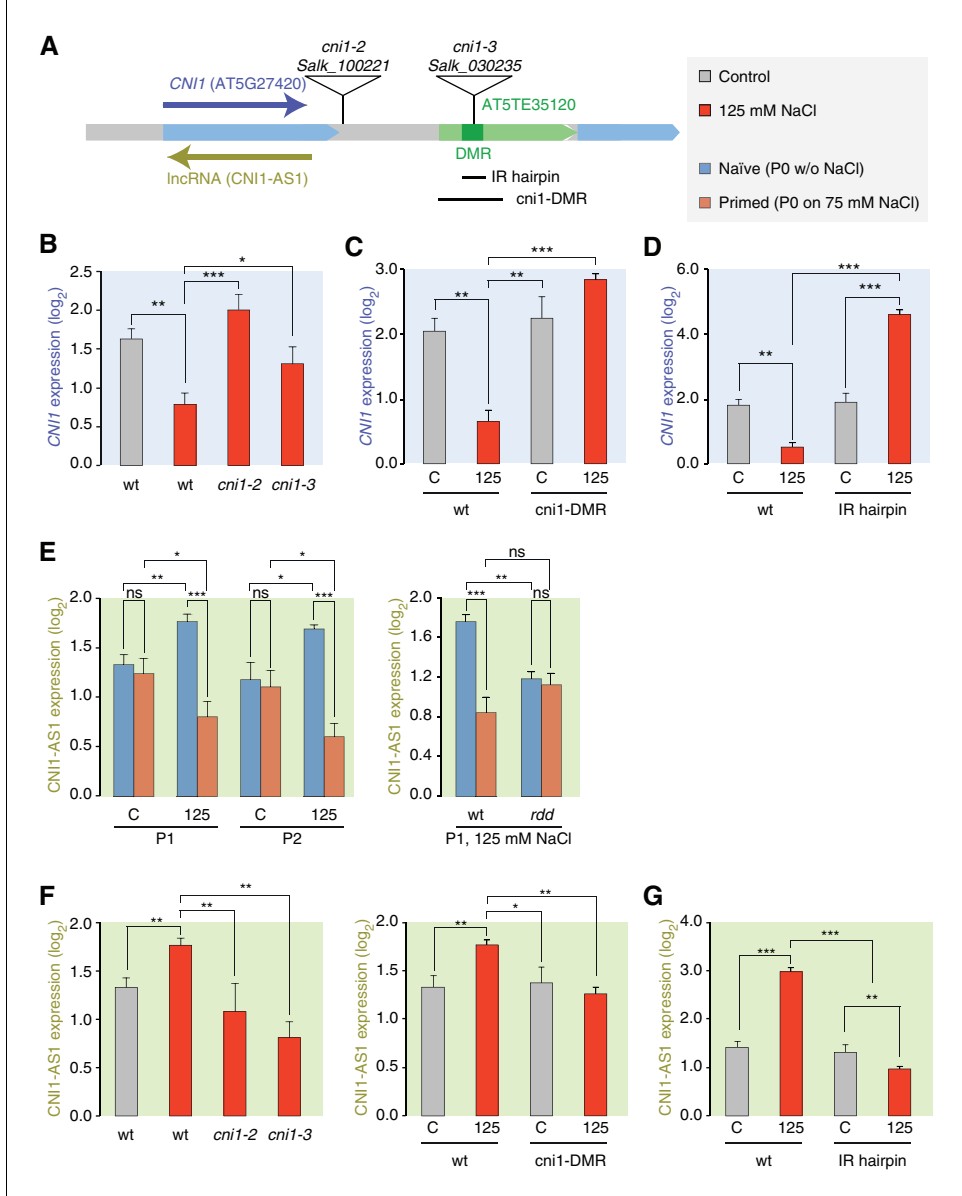

**Figure 6.** lncRNA-mediated control of *CNI1* expression by a salt-induced DMR. (**A**) Diagram of the *CNI1* locus and key for expression experiments. Positions of insertion alleles, *cni1-2* (Salk_100221) and *cni1-3* (Salk_030235), are indicated, as are the DMR (dark green) in the transposable element (AT5TE35120) downstream of *CNI1*, the sequence used for the inverted repeat (IR) hairpin to induce methylation independent of the environment, and the CRISPR/Cas9 created deletion Δ*cni1*-DMR (*cni1*-DRM). (**B, C, D**) Salt-responsive *CNI1* expression in wild type and T-DNA insertion mutants, CRISPR/Cas9 deletion mutants, and IR hairpin transgenic plants. Asterisks indicate significant differences relative to controls (unpaired Student's t-test; *p<0.05, **p<0.01, ***p<0.001, ns p>0.05). Error bars indicate standard deviation. C, control; 125, 125 mM NaCl. (**E**) Salt-responsive lncRNA expression in progeny of naïve or salt-treated wild type and in *rdd* mutants. (**F, G**) Salt-responsive lncRNA expression in wild type and T-DNA insertion mutants, CRISPR/Cas9 mutants, and IR hairpin transgenic plants.

DOI: https://doi.org/10.7554/eLife.13546.023

The following figure supplements are available for figure 6:

**Figure supplement 1.** Generation of CRISPR/Cas9 deletions at the *CNI1* HS-DMR region.
DOI: https://doi.org/10.7554/eLife.13546.024

**Figure supplement 2.** Methylation analysis of hairpin lines directing RdDM hypermethylation at the *CNI1* HS-DMR.
DOI: https://doi.org/10.7554/eLife.13546.025

*Figure 6 continued on next page*

*Figure 6 continued*

**Figure supplement 3.** Expression analysis of CNI1 antisense lncRNA transcripts in response to hyperosmotic stress.

DOI: https://doi.org/10.7554/eLife.13546.026

communication between the HS-DMR and the *CNI1* locus (*Figure 6A*). As expected, *CNI1* sense transcription in both insertion alleles was misregulated in response to stress (*Figure 6B*). To demonstrate the functional relevance of the sequence targeted epigenetically after hyperosmotic-stress, we generated a deletion line (Δcni1-*DMR*) using CRISPR/Cas9 genome editing (*Jinek et al., 2012*) (*Figure 6A*; *Figure 6—figure supplement 1*). Deletion of the HS-DMR sequence reduced downregulation of sense *CNI1* transcripts in response to hyperosmotic stress (*Figure 6C*), indicating that this HS-DMR acts as a distant regulatory element. Finally, to demonstrate directly that the methylation status of this element determines its activity, we introduced an inverted repeat (IR) hairpin (*Matzke et al., 2002*) that directs DNA methylation to the *CNI1* HS-DMR by RdDM (*Figure 6A*). McrBC assays and bisulfite sequencing confirmed that methylation in IR hairpin transgenic lines remained even under hyperosmotic stress (*Figure 6—figure supplement 2*). Levels of *CNI1* sense transcript in response to stress were no longer reduced in these plants (*Figure 6D*), thus strengthening the argument that the *CNI1* HS-DMR acts as an epigenetically sensitive regulatory element.

Because stress can trigger the expression of lncRNAs (*Liu et al., 2012*; *Matsui et al., 2008*), we analysed published datasets (*Jin et al., 2013*) to investigate whether a lncRNA might mediate the effects of the HS-DMR on *CNI1* expression. We found that hyperosmotic stress increased expression of a lncRNA that is transcribed in the antisense direction and overlaps with *CNI1*, both in control plants (*CNI1-AS1*) (*Figure 6—figure supplement 3*) and in the progeny of plants that had gone through our control non-salt conditions (*p*-value 0.004; *Figure 6E*). By contrast, hyperosmotic stress reduced expression of this lncRNA in progeny of plants exposed to hyperosmotic stress (*p*-value 0.015; *Figure 6E*). That this was linked to differential methylation was indicated by the hyperosmotic responsiveness of *CNI1-AS1* being impaired in progeny of *rdd* mutants (*Figure 6E*). In the two insertion alleles and in the deletion line, the stress-mediated transcriptional response was similarly altered (*Figure 6F*). These data support a model in which the HS-DMR downstream of *CNI1* acts as an epigenetic sensor that controls *CNI1* expression by modulating the expression of an antisense lncRNA. Our finding that lncRNA expression was insensitive to hyperosmotic stress in IR hairpin lines (*Figure 6G*) adds further weight to this model.

To investigate whether there might be a broader role for lncRNAs in mediating the effects of stress induced DMRs, we comparatively analysed all stress-responsive genes adjacent to HS-DMRs and to MA-DMRs. Only the first group was enriched for hyperosmotic-responsive antisense lncRNAs (Fisher's Exact test, p=0.008) (*Supplementary file 6*), indicating that HS-DMRs preferentially act as regulatory elements of stress-induced antisense lncRNAs.

## Discussion

The extent and mechanism by which organisms acquire heritable adaptive traits after parental exposure to environmental stress is a central question in genetics and evolution. Unlike animals, plants present a fascinating model to examine this problem because their sessile nature makes it likely that parent and offspring will be exposed to similarly stressful conditions. Here, we have used a systematic approach to assess transgenerational stress adaptation in Arabidopsis (*Figure 1*). Our main conclusion is that intergenerational priming responses to hyperosmotic stress are triggered by recurrent exposure to stimuli, but that this response is rapidly lost in the absence of stress. Whether stress memory is less transient in perennial plants, which are rooted in place over consecutive years, needs to be determined.

In animals that lack DNA methylation, the primary mechanisms involved in the acquisition and inheritance of new characters directed by the environment rely on histone modifications (*Öst et al., 2014*) and small RNAs (*Ashe et al., 2012*; *Shirayama et al., 2012*). In mammals such adaptation is usually associated with changes in DNA methylation (*Radford et al., 2014*). In plants, environmentally-directed heritable traits have also been proposed to be associated with whole-scale changes in global DNA methylation (*Boyko et al., 2010*; *Dowen et al., 2012*; *Jiang et al., 2014*). Bycontrast,

using our highly controlled and replicated syste, and a strict method for detecting methylated regions, we found no clear evidence for such indiscriminate changes. Instead, we found that hyper-osmotic stress directs DNA methylation changes primarily to discrete genome regions that are rich in TE-related sequences (*Figures 2* and *3*). The dynamic DNA methylation changes are of functional consequence because the adaptive response is impaired in mutants defective in DNA methylation pathways (*Figure 1*), which are also known to regulate TE activity (*Kim and Zilberman, 2014*). Mutants defective in DNA methylation and demethylation pathways displayed similar adaptive stress memory abilities, thus indicating that adaptive memory is controlled by complex processes that may not strictly rely on DNA methylation changes alone (*Crisp et al., 2016*).

A small fraction of acquired epigenetic changes is transmitted to the immediate offspring only after recurrent stress exposure, but in the absence of a renewed stimulus these are reset in subsequent generations. This implies that epigenetic stress memory in annual plant species may be transient in nature, however this response may differ in perennial plants that grow over longer periods of time before producing any offspring or in plants that reproduce asexually.

Half of the newly acquired methylation changes identified in our stressed plants overlapped with regions privileged for epimutation also found in near-isogenic greenhouse-grown populations (*Figure 2*) (*Hagmann et al., 2015*). This finding confirms that distinct regions of the plant epigenome are particularly labile, and it suggests that such acquired epimutations could modulate the ability of plants to respond to stress. Importantly, in our system, enhanced stress responses were only passed on after two consecutive generations of stress exposure, but not when plants were exposed to hyperosmotic stress for only one episode, which may be the reason why previous studies often failed to find clear evidence for inheritance of stress-induced epigenetic changes (*Eichten and Springer, 2015*; *Secco et al., 2015*). One explanation for repeated stress exposure being required for induction of transiently heritable stress resistance may reside in the importance of poised epigenetic states for environmental priming (*Jaskiewicz et al., 2011*; *Sani et al., 2013*), which in turn may facilitate the establishment of new epigenetic marks at discrete genomic regions that are sensitive to stress. This view is supported by the substantial overlap (>30%) between HS-DMRs established in recurrently stressed generations (*Figure 2*) and dynamic chromatin occupancy directed by hyperosmosis (*Table 1*). Intriguingly, recurrent exposure to hyperosmotic stress did not significantly increase the number of newly acquired epimutations, suggesting that the extent of epigenetic plasticity elicited by the environment is limited to a few key genomic regions that may be under purifying selection (*Hollister and Gaut, 2009*). It is likely that certain genomic regions are epigenetically targeted, depending on the type of stress because the DNA methylation changes induced by hyperosmotic stress did not overlap with methylation changes reported for other abiotic stresses (*Seymour et al., 2014*). A direct comparison of the latter study to this study is however difficult to make due to inconsistencies in experimental methodology, as *Seymour et al., 2014* did not investigate methylation changes in subsequent generations grown in the absence of stress.

While it is accepted that some epigenetic variation caused by the environment can be transmitted to the immediate offspring of plants and mammals (*Feil and Fraga, 2011*; *Heard and Martienssen, 2014*), the importance of transmission through either the male or female germline has not been previously investigated. Our analyses revealed that hyperosmotic priming responses were transmitted primarily through the maternal germline (*Figure 4*). We attribute this difference to differences in the meiotic transmission of newly acquired DNA methylation marks. In support of this hypothesis, most of the stress-associated methylation changes in leaves were largely absent in mature male gametes (*Figure 4*), indicating that environmentally directed epigenetic marks are more efficiently reset in male rather than in female gametes. Indeed, not only do male gametes undergo an active reprogramming of DNA methylation at transposon sequences by the DNA glycosylase DME (*Borges et al., 2012a*; *Ibarra et al., 2012*), but also at stress-dependent DNA methylation sites (*Figure 4*). Moreover, the enhanced tolerance to hyperosmotic stress in progeny of stressed *dme*-6 plants implicates DNA demethylation both in the active reprogramming of TEs and in the resetting of environmentally-directed epigenetic changes in male gametes. The male transmission of stress-associated adaptive responses is under the strict control of DME (*Figure 4*). Although the precise mechanism of DME in resetting HS-DMRs is unknown, it may be linked to the movement of RNA to sperm cells from the surrounding vegetative cells (*Duan et al., 2016*). Why male and female gametes should differ in their ability to reset newly acquired epigenetic marks remains an enigma. One explanation is that female gametes reside on the mother plant where they are eventually fertilised by sperm from

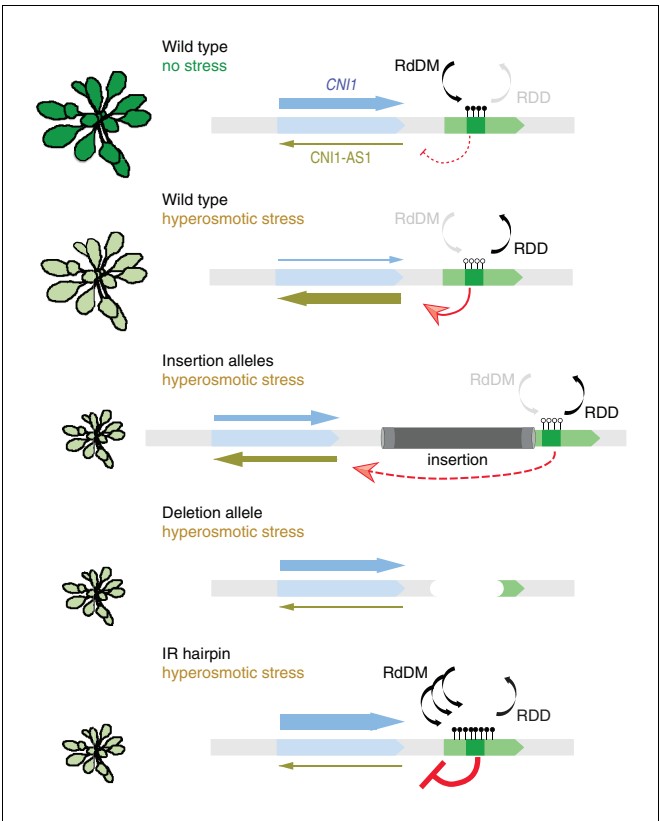

**Figure 7.** Working model for stress-dependent epigenetic regulation of *CNI1*. In wild type, demethylation of a DMR in response to hyperosmotic stress stimulates lncRNA expression (*CNI1-AS1*), which in turn causes downregulation of *CNI1* expression by an unknown mechanism. In insertion and deletion mutants, upregulation of the *CNI1-AS1* is reduced, impairing the salt-dependent reduction in *CNI1* expression. In IR hairpin lines, salt-induced DMR demethylation is countered by forced methylation triggered by the IR hairpin. RdDM, RNA directed DNA methylation activity; RDD, DNA demethylation activity. Black/open lollipops signify methylated/unmethylated cytosines.

DOI: https://doi.org/10.7554/eLife.13546.027

pollen that may have travelled a great distance from its male progenitor to produce seed. Given that seed dispersal usually occurs at a short distance from the mother, a solely maternal transmission of the newly acquired epigenetic marks would be an efficient way of retaining only the most relevant parental stress-associated adaptive responses in the progeny.

Our study has revealed that environmentally-induced priming responses involve DNA (de)methylation pathways that control the extent to which acquired epigenetic states are inherited and maintained in successive generations (*Figure 1*). Previous studies have focused on these pathways because they mediate global stress responses (*Zhu, 2009*). We now provide compelling evidence that stress can specifically alter methylation at the adjacent sequences of several key stress-response regulators (*Figures 2*, *5* and *6*), a process that is mediated by the RdDM and DNA demethylation pathways (*Figure 7*). Targeted studies have revealed that some TEs proximal to upstream regions and sensitive to methylation changes directed by hyperosmotic stress have regulatory roles (*Baek et al., 2011*; *Xu et al., 2015*) and are associated with quantitative traits and adaptive behaviour (*Baxter et al., 2012*; *Busoms et al., 2015*). Notably, both DNA methylation and demethylation activities are also tightly regulated by the action of adjacent upstream TEs that act as an epigenetic sensors (*Lei et al., 2015*; *Williams et al., 2015*). The epigenetic regulation of HS-DMRs is not always associated with upstream regulatory sequences and sense transcription; instead we found that they are preferentially associated with the transcription of antisense lncRNAs (*Figure 6*). Although the functions of these HS-DMR-associated non-coding transcripts are largely unknown, antisense lncRNAs have been implicated in directing chromatin changes (*Ariel et al., 2014*; *Heo and Sung,*

2011; Swiezewski et al., 2009), and thereby in influencing transcription, splicing and transcript stability (Bardou et al., 2014; Borsani et al., 2005). In this study, antisense lncRNAs were shown to be regulated by epigenetically labile control elements sensitive to stress (Figure 7). Methylation changes in these genome regions could modulate transcription factor binding (Zhong et al., 2013) or chromatin regulatory loops (Ariel et al., 2014).

Adaptive epigenetic inheritance has been a topic of fascination, but also of scientific controversy (Lysenko, 1951). The adaptive value of this inheritance over multiple generations must depend on the cost of epigenetic resetting, as well as on the degree and predictability of environmental stress (Herman et al., 2014). Because conditions in many natural environments are highly stochastic, an adaptive bet-hedging strategy (Simons, 2011) that is mediated by increased epigenetic variation could be advantageous. Under our controlled stress conditions, the contribution of adaptive epigenetic variation shows parental effects (Figure 4), which may be favoured over bet-hedging in relatively stable environments (Kuijper and Johnstone, 2016). Hence, our work provides insights into the importance of epigenetically driven adaptive changes and illustrates the evolutionary significance of epigenetic plasticity in plants.

## Materials and methods

### Plant growth and material
The wild-type background studied was *A. thaliana* Col-0. The multigenerational hyperosmotic stress experiments used the reporter line L5, which harbours a silenced reporter encoding β-glucuronidase linked to the cauliflower mosaic virus 35S promoter (35Spro::GUS) (Morel et al., 2000). To isolate male gametes, we used reporter lines *MGH3p::MGH3-eGFP* and *ACT11p::H2B-mRFP*, in which either sperm or vegetative cells are marked by fluorescent protein expression (Borges et al., 2012b) (Figure 4—figure supplement 1). Mutants *cmt3-11* (Chan et al., 2006 *drm1-2 drm2-2* (Chan et al., 2006), *dme*-6 (Shirzadi et al., 2011), *nrpda1-4* (Herr et al., 2005), *rdr2-1* (Xie et al., 2004), *ros1-4* (Zheng et al., 2010), and the *ros1-3 dml2-1 dml3-1* (Penterman et al., 2007) triple mutant have been described. Plants were grown at 22°C under long days (16 hr light, 8 hr dark; light intensity 120 µmol/sec/m$^2$). Lines carrying T-DNA insertions downstream of *CNI1* were obtained from the SALK collection (*cni1-2*, Salk_100221 and *cni1-3*, Salk_030235).

### Multigenerational salt treatments
Seeds from a single founder plant were germinated and grown on MS medium (control) for two weeks and transferred to MS medium supplemented with 25 or 75 mM NaCl to induce mild hyperosmotic stress for 4 weeks. Before flower buds were visible, plants were transferred to soil (generation 1). We sampled 10 individual plants from each treatment, whereby ten-week-old leaf samples and mature seeds were collected separately from each plant. This process was repeated for five successive generations. In each generation, offspring of the salt treated and control plants were grown in non-stress condition for two successive generations to produce P1 and P2 plants (Figure 1A).

### Generation of CRISPR/Cas9 lines for the deletion of HS-DMR
We modified plasmids previously described (Fauser et al., 2014). A pair of guide RNAs was selected using the CRISPR-PLANT tool (Xie et al., 2014), the corresponding DNA oligonucleotides (Integrated DNA Technologies) were cloned into pEN-Chimera using BbsI and BsmBI to generate plasmids pEN-CNI1.1. Constructs were transferred into pDE-CAS9 plasmid by Gateway cloning (Invitrogen) and transformed by floral dipping (Clough and Bent, 1998). Deletions where identified by PCR (Supplementary file 7) and confirmed by sequencing.

### IR hairpin lines
A DNA fragment was chemically synthesized (IDT) and introduced into the hairpin vector pJawohl-Act2 using Gateway cloning (Life Technologies). Constructs were transformed by floral dipping. Transgenic T2 lines (T2) with DNA hypermethylation at the *CNI* HS-DMR after exposure to hyperosmotic stress (175 mM NaCl) were identified by CHOP-PCR (Zhang et al., 2014) after digestion with HpyCH4IV (NEB) and PCR amplification (Supplementary file 7).

## Germination and survival test

All phenotypic tests were carried out with six independent replicates. For germination assays, 50 seeds were sown per plate on MS with or without 200 mM NaCl, a concentration of salt we found to be highly selective in the germination of Col-0 seeds. Seeds were scored as having germinated based on radicle emergence 14 days after sowing. For survival assays, 50 seeds were sown on MS or on MS supplemented with 150 mM NaCl, a concentration of salt known to allow germination but affecting vegetative growth in Col-0. Survival was scored based on presence/absence of green leaves 14 days after sowing. Data are summarized in *Supplementary file 1*.

## Chlorophyll content assay

Plants were grown on MS medium with or without 100 mM NaCl for 5 weeks. Leaves were collected, weighed, and washed in distilled water. Chlorophyll was extracted by incubating 0.02 – 0.03 g of ground leaf material in 80% (v/v) aqueous acetone at 4℃ for 48 hr. Total chlorophyll content (chlorophyll a and b) was measured using a spectrophotometer at 663.6 nm and 646.6 nm absorbance (*Porra, 2002*).

## Sodium content assay

Plants were grown on MS medium with or without 100 mM NaCl for 5 weeks. Leaves were collected, and washed in distilled water. Plant material was dried at 80℃ for 48 h, then weighed. Ions were acid-extracted from dried plant material using 2 ml of concentrated nitric acid and microwave digestion. The digestion program consisted of: 5 min at 100℃, 2 min at 120℃, 5 min on 160℃, 22 min at 180℃, and cooling down to 70℃. After samples had cooled down, the digested samples were diluted with 23 ml distilled water. The sodium ion concentration of the diluted samples was measured using Inductively Coupled Plasma Mass Spectrometry (ICP-MS).

## Isolation of sperm cell and vegetative nuclei

*MGH3::MGH3-eGFP/ACT11::H2B* plants were germinated and grown for 6 weeks on MS medium without NaCl or with 25 mM or 75 mM NaCl before being transferred to soil to induce flowering. Approximately 10 g of flower tissue were collected into 50 ml Falcon tubes, 10 ml of sperm nuclei buffer was added and the pollen suspension was vortexed for 3 min. The pollen suspension was filtered through a Miracloth mesh and centrifuged for 1 min at 3000 rpm; the supernatant was carefully removed from the pollen pellet. For the extraction of nuclei, the pellet was re-suspended in 1 ml sperm nuclei buffer, loaded into 1.5 ml tubes containing 100 µl of acid-washed glass beads (425–600 µm) and mixed for 4 min. The crude extract was filtered through a 28 µm micro-filter sieve, leaving the nuclei intact. Vegetative and sperm nuclei were isolated from the crude extract of disrupted pollen using Fluorescence-Activated Cell Sorting (FACS) with a MoFlo high-speed cell sorter (Beckman Coulter, Fort Collins, USA) (*Borges et al., 2012b*) (*Figure 4—figure supplement 1*). One laser was set to 140 mW at 488 nm for forward scatter (FSC) and side scatter (SSC) measurements, and for GFP excitation. A second laser was set to 38 mW at 561 nm for RFP excitation. GFP and RFP were detected using 530/40 nm and 630/75 nm bandpass filters.

## RNA analysis

P1 and P2 progeny of the G3 generation were grown on MS medium supplemented with 125 mM NaCl for 2 weeks. Leaves were collected from 50 seedlings and total RNA was extracted using the RNeasy Plant Mini Kit (Qiagen) according to the manufacturer's instructions. RNA was treated with TURBO DNA-free (Promega, Madison, WI). cDNA was synthesized from 1 µg of extracted RNA using the RevertAid First Strand cDNA Synthesis Kit (Thermo Scientific). Quantitative real time PCR analyses were performed on a MyiQ System (BIO-RAD), using oligonucleotide primers designed with Primer3 (*Rozen and Skaletsky, 2000*) (*Supplementary file 7*). PCR fragments were analysed using a dissociation protocol to ensure that each amplicon was a single product. Amplicons were also sequenced to verify the specificity of PCR. The amplification efficiency was calculated from raw data using LingRegPCR (*Ramakers et al., 2003*). All RT-qPCR experiments were performed using five biological replicates, with a final volume of 25 µl containing 5 µl of cDNA template (diluted beforehand 1:10), 0.2 µM of each primer, and 12.5 µl of 2×MESA Blue qPCR MasterMix (Eurogentec Headquarters). The following cycling profile was used: 95℃ for 10 min, followed by 40 cycles of

95°C for 10 s, 60°C for 15 s, and 72°C for 15 s. The melting curve was determined in the range of 60–95°C, with a temperature increment of 0.01°C/sec. Each reaction was run in triplicate (technical replicates). Negative controls included in each run were a reaction without reverse transcriptase and one without template (2 µL of nuclease-free water instead of 2 µL of cDNA). No signals were observed in the negative controls. Raw Ct data were analysed using GeneEx Pro (*Kubista et al., 2006*). Analysis of expression data was performed according to the ΔΔCT method (*Livak and Schmittgen, 2001*) using *GADPH* (At1g13440), *PDF2* (At1g13320) and *UBQ5* (At3g62250) for normalization (*Lippold et al., 2009*). To measure *CNI1* antisense lncRNA transcripts, 1 µg of total RNA was isolated from seedlings. Reverse transcription used At5g27420_ant, PP2AA3 Reverse and GAPDH Reverse oligonucleotides (*Supplementary file 7*) in the same reaction with SuperScriptIII Reverse Transcriptase (Invitrogen). qPCR reactions used At5g27420 Forward and At5g27420 Reverse primers (*Supplementary file 7*) following the same conditions described for the sense reactions. These experiments were performed using six technical replicates for each reaction. Expression data were analysed according to the ΔΔCT method (*Livak and Schmittgen, 2001*) using *GADPH* (At1g13440) and *PP2AA3* (At1g13320) for normalization. PCR reactions were performed in duplicate and RT-minus controls were included to confirm absence of genomic DNA contamination.

## Bisulfite sequencing

For somatic tissue, rosette leaves were pooled from 10 plants for each treatment group. For male gamete analysis, sperm and vegetative nuclei were collected from 100 plants for each treatment group. gDNA was extracted from leaf samples with the DNAeasy Plant Mini Kit (Qiagen), and from sperm and vegetative nuclei with MasterPureDNA Purification Kit (Epicentre). DNA libraries were generated using the Illumina TruSeq Nano kit (Illumina, CA, USA). DNA was sheared to 350 bp. The bisulfite treatment step using the Epitect Plus DNA Bisulfite Conversion Kit (Qiagen, Hilden, Germany) was inserted after the adaptor ligation; incubation in the thermal cycler was repeated once before clean-up. After clean-up of the bisulfite conversion reaction, library enrichment was done using Kapa Hifi Uracil+ DNA polymerase (Kapa Biosystems, MA, USA). Libraries were sequenced with 2 x 101 bp paired-end reads on an Illumina HiSeq 2000 instrument, with conventional gDNA libraries in control lanes for base calling calibration. Seven to eight libraries with different indexing adapters were pooled in one lane. For image analysis we used Illumina RTA 1.13.48.

## Processing and alignment of bisulfite-converted reads

The procedure followed (*Becker et al., 2011*). In brief, the SHORE pipeline v0.9.0 (*Ossowski et al., 2008*) was used to trim and quality-filter the reads. Reads with more than 5 (or 2) bases in the first 25 (or 12) positions with a base quality score of below 5 were discarded. Reads were trimmed to the right-most occurrence of two adjacent bases with quality values of at least 5. Trimmed reads shorter than 40 bases were discarded. Reads were then aligned against the Col-0 reference genome sequence using GenomeMapper implemented in SHORE (*Schneeberger et al., 2009*).

## Identification of methylated sites and differentially methylated positions (DMPs)

We used published methods (*Becker et al., 2011*). The number of covered and methylated sites for each sample as well as the false methylation frequencies retrieved from read mappings against the chloroplast sequence can be found in *Supplementary file 1*. On average, 40.7 million cytosines were covered by at least 3 reads and with a quality score above 25 in more than half of the samples. Of these, 7.2 million cytosines were methylated in at least one sample (*Supplementary file 8*). For DMP calling, we modified the approach from *Becker et al. (2011)*, without removing sites classified as differentially methylated between replicates. We applied Fisher's Exact test for all pairwise sample comparisons on cytosine sites with a methylation frequency difference to another sample of at least 30%. We used the same *P* value correction scheme as in *Becker et al. (2011)*.

## Identification of methylated regions (MRs) and differentially methylated regions (DMRs)

We first identified MRs in each sample separately using a Hidden Markov Model (HMM) (*Hagmann et al., 2015*). MRs of replicates were merged into a common set of MRs. Whenever

different samples were treated as a replicate group (e.g. control and salt-treated samples), their MRs were merged into a common set. Regions that showed statistically significant methylation differences between at least two sets of strains were identified as DMRs (*Hagmann et al., 2015*). In brief, segmentations across the genomes of every sample served to set breakpoints of start and end coordinates of all predicted MRs. Each combination of coordinates in this set defined a segment to perform the test for differential methylation in all pairwise comparisons of the strains, if at least one strain was in a high methylation state throughout this whole segment (*Hagmann et al., 2015*). Per pairwise comparison, between 30,000 and 50,000 segments were tested (*Hagmann et al., 2015*). For tests within generations, we grouped P0 control, P1 control and P2 control samples as 'non-stressed'; P0 salt-treated samples as 'stressed'; P1 samples derived from salt-treated P0 plants as 'stressed-P1'; and P2 samples derived from salt-treated P0 plants as 'stressed-P2'. Tests for DMRs were then carried out between these four groups. In addition we separately tested without the respective remaining groups for 'non-stressed' vs. 'stressed', 'stressed' vs. 'stressed P1', 'stressed' vs. 'stressed P2', and 'stressed P1' vs 'stressed P2'. This latter step was done to assess the number of DMRs directly identified between two groups, without multiple testing correction for comparisons with and between other groups. DMRs from the MA lines were taken from a previous publication (*Hagmann et al., 2015*).

## Mapping to genomic elements

We used the TAIR10 annotation for genes, exons, introns and untranslated regions; transposon annotation was according to *Slotte et al. (2013)*. Positions and regions were hierarchically assigned to annotated elements in the order CDS > intron > 5' UTR > 3' UTR > transposon > intergenic space. We defined as intergenic positions and regions those that were not annotated either as CDS, intron, UTR or transposon. Each position was assigned to the corresponding element that contained it. DMRs were assigned to annotated elements by basepair, i.e. each position in the DMR was assigned in the above-mentioned order. A DMR can stretch over several annotated elements. We tested the overlap of DMRs with other DMRs or with genes using bedtools (*Quinlan and Hall, 2010*), either requesting a direct overlap or an overlap within a window of *n* bp downstream and upstream of the regions. For overlap between DMRs and TEs, we required either a direct overlap or an overlap within 2,000 bp windows downstream and upstream of the DMRs. Overlapping TEs were then sorted into their superfamilies according to TAIR nomenclature. The TE profiles for hypo- and hypermethylated DMRs were compared against the expected values taken from the whole genome TE profile. For each TE superfamily the expected values were calculated as: $Se = (wgs/wgt) * st$, where *Se* is the expected value for that superfamily in that sample (hypo or hyper), *wgs* is the number of transposons of that superfamily in the whole genome, *wgt* is the total number of transposons in the whole genome, and *st* is the total number of transposons in this sample.

## Gene expression and ontology analysis

For the identification of genes regulated by HS-DMRs we first identified the genes within 2 kb upstream or downstream of DMRs and analysed their expression in shoots or roots exposed to hyperosmotic stress using AtGenExpress (*Kilian et al., 2007*). We used Protein ANalysis THrough Evolutionary Relationships (PANTHER 9.0) software (*Mi et al., 2013*) to classify significantly enriched Gene Ontology (GO) terms associated with/without overlap with MA line DMRs and with hypo- and hypermethylated DMRs. Heatmaps for GO analysis were generated using R version 3.0.1 (www.r-project.org).

## Statistical analysis

Complete linkage clustering was done in R version 3.0.1 (www.r-project.org) using the 'heatmap.2' function of the 'gplots' package in combination with the 'hclust' function of the 'fastcluster' package using the complete linkage clustering method. Uncertainty in hierarchical clustering analyses was estimated using the *pvclust* package in R.

## Data visualization

Graphs were generated using R version 3.0.1 (www.r-project.org). Circular display of genomic information in chromosomes was rendered using Circos version 0.63 (*Krzywinski et al., 2009*).

## Data accessibility

The DNA and RNA sequencing data have been deposited at the European Nucleotide Archive under accession numbers PRJEB9076 and PRJEB13558. DNA methylation data and MR coordinates have been uploaded to the epigenome browser of the EPIC Consortium (https://www.plant-epigenome. org/; https://genomevolution.org/wiki/index.php/EPIC-CoGe) and can be accessed at http://genomevolution.org/r/939v .

## Acknowledgements

We thank L Costa, as well as D Seymour and other members of the Weigel lab for discussions and critical reading of the manuscript, C Lanz for help with Illumina sequencing, and S Easterlow, H Lanang Putra, Q Saintain and F Nungky Harjanti for technical support. This work was supported by the Royal Society, ESF/RTD Framework COST action (FA0903), BBSRC (BB/F008082), DFG (SFB 1101 - Project C01), the Max Planck Society, the EU FP7 Collaborative Project Grant AENEAS, and the ERA-CAPs project EVOREPRO.

## Additional information

### Competing interests

Detlef Weigel: Senior editor, *eLife*. The other author declares that no competing interests exist.

### Funding

| Funder | Grant reference number | Author |
| --- | --- | --- |
| Max-Planck-Gesellschaft | | Detlef Weigel<br>Jorge Kageyama<br>Claude Becker |
| Deutsche Forschungsge-meinschaft | SFB 1101- Project C01 | Detlef Weigel |
| ESF/RTD Framework COST action | FA0903 | Jose Gutierrez-Marcos |
| Biotechnology and Biological Sciences Research Council | BB/F008082 | Jose Gutierrez-Marcos |
| EU FP7 Collaborative Project Grant | AENEAS | Jose Gutierrez-Marcos |
| ERA-CAPS Project | EVOREPRO | Jose Gutierrez-Marcos |
| The Royal Society | IE150496 | Jose Gutierrez-Marcos |

The funders had no role in study design, data collection and interpretation, or the decision to submit the work for publication.

### Author contributions

Anjar Wibowo, Claude Becker, Acquisition of data, Analysis and interpretation of data, Drafting or revising the article; Gianpiero Marconi, Acquisition of data, Analysis and interpretation of data; Julius Durr, Designed strategy for CRISPR/Cas9 deletion lines, Acquisition of data, Analysis and interpretation of data; Jonathan Price, Analysis of DNA methylation and gene expression data, Performed the computational analysis; Jorg Hagmann, Analysis of DNA methylation data, Performed the computational analysis; Ranjith Papareddy, Performed the experiments, Analysis and interpretation of data; Hadi Putra, Quantitative PCR expression analysis, Analysis and interpretation of data; Jorge Kageyama, Performed the computational analysis, Acquisition of data; Jorg Becker, Isolation of Sperm cell and Vegetative nuclei for DNA methylation analysis, Performed the FACS experiments; Detlef Weigel, Wrote the manuscript, Conception and design; Jose Gutierrez-Marcos, Conception and design, Analysis and interpretation of data, Drafting or revising the article

## Author ORCIDs

Claude Becker (iD) http://orcid.org/0000-0003-3406-4670
Gianpiero Marconi (iD) http://orcid.org/0000-0003-2669-0399
Detlef Weigel (iD) http://orcid.org/0000-0002-2114-7963
Jose Gutierrez-Marcos (iD) http://orcid.org/0000-0002-5441-9080

## Decision letter and Author response

Decision letter https://doi.org/10.7554/eLife.13546.043
Author response https://doi.org/10.7554/eLife.13546.044

# Additional files

## Supplementary files

• Supplementary file 1. Phenotypic data.
DOI: https://doi.org/10.7554/eLife.13546.028

• Supplementary file 2. Methylation sequencing statistics.
DOI: https://doi.org/10.7554/eLife.13546.029

• Supplementary file 3. Differentially methylated positions (DMPs).
DOI: https://doi.org/10.7554/eLife.13546.030

• Supplementary file 4. Differentially methylated regions (DMRs).
DOI: https://doi.org/10.7554/eLife.13546.031

• Supplementary file 5. Genes in proximity of HS-DMRs.
DOI: https://doi.org/10.7554/eLife.13546.032

• Supplementary file 6. Intersection between stress-induced lncRNAs and HS-DRM associated genes.
DOI: https://doi.org/10.7554/eLife.13546.033

• Supplementary file 7. List of oligonucleotides employed in PCR analyses.
DOI: https://doi.org/10.7554/eLife.13546.034

• Supplementary file 8. Scoring matrix for assessing cytosine site statistics.
DOI: https://doi.org/10.7554/eLife.13546.035

## Data availability

The following datasets were generated:

| Author(s) | Year | Dataset title | Dataset URL | Database and Identifier |
|---|---|---|---|---|
| Becker C | 2015 | Exposure to environmental stress induces a transient epigenetic memory response in plants | http://www.ebi.ac.uk/ena/data/search?query=PRJEB9076 | European Nucleotide Archive, PRJEB9076 |
| Becker C | 2016 | Exposure to environmental stress induces a transient epigenetic memory response in plants | http://www.ebi.ac.uk/ena/data/search?query=PRJEB13558 | European Nucleotide Archive, PRJEB13558 |

The following previously published dataset was used:

| Author(s) | Year | Dataset title | Dataset URL | Database and Identifier |
|---|---|---|---|---|
| Jiang Caifu | 2014 | Arabidopsis thaliana Col-0 BS data: G11-C1 | http://www.ncbi.nlm.nih.gov/sra/SRX703644[accn] | NCBI Sequence Read Archive, SRP047267 |

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
