## [Decision Letter]

Thank you for submitting your work entitled "Stress memory in *Arabidopsis* is mediated by discrete epigenetically labile genome regions" for consideration by *eLife*. Your article has been favorably evaluated by Ian Baldwin (Senior editor) and three reviewers, one of whom, Sheila McCormick, is a member of our Board of Reviewing Editors.

The reviewers have discussed the reviews with one another and the Reviewing Editor has drafted this decision to help you prepare a revised submission.

Summary:

This manuscript investigates the potential for environmental stress, in this case hyperosmotic treatment, to cause heritable changes in phenotypes via epigenetic information, in this case DNA cytosine methylation, using the model plant *Arabidopsis thaliana*. The role of epigenetic information in mediating such forms of 'soft inheritance' is of great general interest, though relatively few clear examples have been discovered so far in plants. Indeed, although several recent reports concluded that there is epigenome-mediated transgenerational memory of stress in plants, many of these studies were not rigorous in terms of design or analysis, or focused on effects at complex transgenic loci, the applicability of which to endogenous genes is unclear. By contrast, this study is carefully designed and analyzed. Detailed genetic experiments at the CNI locus support take home messages from the whole genome experiments. Overall, this is an impressive paper but certain key aspects need further clarification and/or experiments.

Essential revisions:

1) The key phenotype observed is reported in Figure 1B; Col-0 plants were grown under control or hyperosmotic stress for five generations, then unstressed P1 or P2 generation progeny were tested for salt tolerance to assess transgenerational inheritance of stress adaptation. The assay used is survival rate of progeny seedlings grown on salt. Example survival plates should be shown to provide clear phenotypic evidence of the stress adaptation. Please also replot the figures in 1B (and Figure 1—figure supplement 1) with the same Y-axis values, ideally 0-100%, which would make it easier to compare plots. A clear increase in progeny survival is seen in P1 generations, with effects lost in P2.

2) There are actually very few stress-induced DNA methylation changes that are transmitted to progeny (i.e., 0 HS-DMRs overlap between all three generations, and only 35 are present in two generations – Figure 2D). This should be made clearer in the summary and/or title, instead of focusing solely on the exceptional cases of inheritance of stress-induced methylation changes.

3) These phenotypes were then tested in a variety of epigenetic mutants that influence DNA methylation establishment, maintenance or removal. A subset are shown in Figure 1, with a greater number in Figure 1—figure supplement 1. As showing genetic dependency of these effects makes this very convincing we suggest that the complete mutant set be shown in Figure 1. Some of the mutants have different survival phenotypes in the control treatments – were statistical tests performed for these comparisons, e.g. Col vs. *cmt3*?

4) The authors then perform reciprocal crossing experiments to show that changes in methylation are differential via the male and female germ line; i.e. transgenerational adaptation is more strong in maternal than paternal crosses, but they further show that stress memory can be transmitted paternally if *dme* is mutated, and that there are almost no methylation differences between the VN and SC of control and hyperosmotic-treated plants. The implication is that the action of DME in the VN is responsible for erasing acquired methylation marks in wild type plants. Ideally this would be supported by data showing increased methylation at HS-DMR regions in the sperm cells of stressed *dme* plants. In the absence of these data the authors should compare the HS-DMRs with *dme* DMRs defined by Ibarra et al. Is there significant overlap?

5) It's surprising that both RdDM establishment mutants (*drm2, rdr2, nrpd1a*) and non-CG maintenance mutants (*cmt3*) and demethylation mutants (*rdd, ros1*) have the same phenotype with regard to survival under stress conditions (Figure 1 – progeny of stressed plants show no enhanced survival) given their opposing molecular functions. This could potentially be explained by reduced ROS1 activity in RdDM mutants, but ROS1 expression is not reduced in *cmt3*. The authors must reconcile or provide possible explanations for how these mutants, which would have quite different/opposite effects on DNA methylation, cause the same abolition of transgenerational stress inheritance.

6) To connect changes between DNA methylation triggered by stress and specific genes the authors then looked for genes that were in proximity to DMRs and which exhibit changes in transcription during stress treatment. Two such genes are analysed in detail. *MYB20* which shows hypermethylation caused by salt stress, and *CNI1* which is hypomethylated. Mutants are analysed, the results again somewhat confusing, as for *MYB20* the *rdd* and RdDM mutants both show reduced expression, yet should be having opposite effects on DNA methylation. An impressive experiment is performed using Cas9 that eliminates that *CNI1* DMR and alters its regulation in response to stress – perhaps *CNI1* should be made more of a focus in the manuscript and certainly should be described in the Abstract.

7) An extensive bisulfite sequencing experiment is then performed on the Col samples and differential methylation analysis performed both for individual cytosines (DMPs) and regions (DMRs). This section was quite hard to follow and to extract the 'take home message'. Statements like 'small but noticeable effect' and 'discovered our DMR regions were present.… but they were below our statistical cut-off' do not give the reader confidence. Surely, the point of a cut-off is that regions below that should not be considered? Is it worth reporting such effects, which seem more or less anecdotal? The complexity of this section is compounded by the fact that stochastic methylation change is observed between generations, as reported by the Weigel and Ecker labs. However, the authors do a good job of disentangling these effects from those caused by the stress. Gain of non-CG methylation at DMRs is the strongest effect caused by the salt stress, so that should be stated more clearly and earlier. The DMP analysis does not seem that relevant, as there is very little evidence that single methylation polymorphisms are biologically meaningful, and thus could be deleted from the manuscript.

8) Correlations are made between changes in DNA methylation and H3K27 methylation seen in a previously published study. How comparable are the stress treatments between that study and this, in addition to the tissues profiled? It is not clear what mechanistic relationship is being proposed. Is it possible to test P1/P2 transgenerational stress responses in mutants in the H3K27 pathway? This section didn't really add much to an already complex and data-rich paper.

9) Throughout the text it needs to be made clearer what experiments the authors performed and what experiments are from published data and were not performed under the same growth regime. Examples of this include the chromatin profiling from hyperosmotic stressed plants (Table 1), and transcriptional profiling upon hyperosmotic treatment (Figure 5—figure supplement 1). Furthermore, the reasons why the salt treatments vary for different experiments should be explained – i.e. 25 or 75 mM for the initial experiments, for 6 weeks, but then 200 mM for the germination test but 150 mM for the survival test (for 14 days each), Chlorophyll content used 100 mM for 5 weeks, and the RNA analysis used 125 mM for 5 weeks. The authors might also mention which, if any, of these levels are considered reasonable for salt stress in nature.

10) Certain classes of transposons are identified as being enriched in DMRs associated with salt treatment. The locus-specific blow ups in Figure 5A were convincing, but the authors should consider providing some other specific loci and transposons that serve as representatives of the changes observed, alongside the summary graphs.

11) In Figure 2B, gain of non-CG methylation appears to be the strongest effect. Is the model that this is stress triggered de novo DNA methylation? The authors should consider repeating part of this experiment in a *drm1drm2* mutant, which should abolish these changes if they are mediated via the canonical RdDM pathway. Demonstrating the genetic dependence of this effect on de novo methylation would strengthen the argument.

---

## [Author Response]

Essential revisions:

1) The key phenotype observed is reported in Figure 1B; Col-0 plants were grown under control or hyperosmotic stress for five generations, then unstressed P1 or P2 generation progeny were tested for salt tolerance to assess transgenerational inheritance of stress adaptation. The assay used is survival rate of progeny seedlings grown on salt. Example survival plates should be shown to provide clear phenotypic evidence of the stress adaptation.

We have included examples of survival plates in Figure 1B.

Please also replot the figures in 1B (and Figure 1—figure supplement 1) with the same Y-axis values, ideally 0-100%, which would make it easier to compare plots. A clear increase in progeny survival is seen in P1 generations, with effects lost in P2.

We have re-plotted all bar charts in Figure 1 and in associated Figure 1—figure supplement 1 as suggested.

2) There are actually very few stress-induced DNA methylation changes that are transmitted to progeny (i.e., 0 HS-DMRs overlap between all three generations, and only 35 are present in two generations – Figure 2D). This should be made clearer in the summary and/or title, instead of focusing solely on the exceptional cases of inheritance of stress-induced methylation changes.

We have made clear in the text that the inheritance of stress-induced methylation changes is rare. Specifically, in title and Abstract, we have added “a limited set of regions”.

3) These phenotypes were then tested in a variety of epigenetic mutants that influence DNA methylation establishment, maintenance or removal. A subset are shown in Figure 1, with a greater number in Figure 1—figure supplement 1. As showing genetic dependency of these effects makes this very convincing we suggest that the complete mutant set be shown in Figure 1.

We have now included all the data for the different mutants tested in Figure 1.

Some of the mutants have different survival phenotypes in the control treatments – were statistical tests performed for these comparisons, e.g. Col vs. cmt3?

Because we are focusing on assessing only the transmission of adaptive stress responses in different epigenetic mutant backgrounds, we performed statistical tests only between treated versus untreated plants of the same genetic background.

*4) The authors then perform reciprocal crossing experiments to show that changes in methylation are differential via the male and female germ line; i.e. transgenerational adaptation is more strong in maternal than paternal crosses, but they further show that stress memory can be transmitted paternally if dme is mutated, and that there are almost no methylation differences between the VN and SC of control and hyperosmotic-treated plants. The implication is that the action of DME in the VN is responsible for erasing acquired methylation marks in wild type plants. Ideally this would be supported by data showing increased methylation at HS-DMR regions in the sperm cells of stressed dme plants.*

We agree that this experiment would be very informative, but it would take two consecutive generations of hyperosmotic stress to generate this material. In addition, because *dme-6* plants can only be grown in heterozygosis, pollen samples will contain a mixture of wild-type and mutant sperm cells and vegetative nuclei, which will make the methylation analysis challenging. We hope that our results are sufficiently convincing even without these additional data, which we agree would be very interesting.

*In the absence of these data the authors should compare the HS-DMRs with dme DMRs defined by Ibarra et al. Is there significant overlap?*

As suggested by the reviewers, we have re-analyzed the *dme* BS-seq data from Ibarra et al. 2012. We compared DMRs present in *dme* sperm and vegetative nuclei to HS-DMRs and as control we also compared to non-DMRs. We found that both HS-DMRs and non-DMRs were similarly affected in *dme* pollen, indicating that genomic loci affected by salt are only a fraction of the total loci that are under the control of DME in the male germ line. These results are now displayed in the new Figure 4—figure supplement 2.

5) It's surprising that both RdDM establishment mutants (drm2, rdr2, nrpd1a) and non-CG maintenance mutants (cmt3) and demethylation mutants (rdd, ros1) have the same phenotype with regard to survival under stress conditions (Figure 1 – progeny of stressed plants show no enhanced survival) given their opposing molecular functions. This could potentially be explained by reduced ROS1 activity in RdDM mutants, but ROS1 expression is not reduced in cmt3. The authors must reconcile or provide possible explanations for how these mutants, which would have quite different/opposite effects on DNA methylation, cause the same abolition of transgenerational stress inheritance.

Adaptation to abiotic stress is a complex trait that is not only regulated epigenetically, but involves many other factors (e.g. immediate transcriptional responses, RNA stability, splicing, metabolites) (Crisp et al. 2016). Because we do not precisely know the effects that the different DNA methylation mutants tested have on these other factors, it is difficult to explain fully the adaptive phenotypes observed. We have included these points in the manuscript as suggested by the reviewers.

*6) To connect changes between DNA methylation triggered by stress and specific genes the authors then looked for genes that were in proximity to DMRs and which exhibit changes in transcription during stress treatment. Two such genes are analysed in detail. MYB20 which shows hypermethylation caused by salt stress, and CNI1 which is hypomethylated. Mutants are analysed, the results again somewhat confusing, as for MYB20 the rdd and RdDM mutants both show reduced expression, yet should be having opposite effects on DNA methylation. An impressive experiment is performed using Cas9 that eliminates that CNI1 DMR and alters its regulation in response to stress – perhaps CNI1 should be made more of a focus in the manuscript and certainly should be described in the Abstract.*

We have included *CNI1* in the Abstract and described it in more detail in the text.

7) An extensive bisulfite sequencing experiment is then performed on the Col samples and differential methylation analysis performed both for individual cytosines (DMPs) and regions (DMRs). This section was quite hard to follow and to extract the 'take home message'. Statements like 'small but noticeable effect' and 'discovered our DMR regions were present.… but they were below our statistical cut-off' do not give the reader confidence. Surely, the point of a cut-off is that regions below that should not be considered? Is it worth reporting such effects, which seem more or less anecdotal? The complexity of this section is compounded by the fact that stochastic methylation change is observed between generations, as reported by the Weigel and Ecker labs. However, the authors do a good job of disentangling these effects from those caused by the stress. Gain of non-CG methylation at DMRs is the strongest effect caused by the salt stress, so that should be stated more clearly and earlier.

We have stated that non-‐CG methylation is strongly affected later in the text because previous studies have concluded that most methylation changes induced by hyperosmotic stress affect CG methylation. We have modified this section to make this point clearer.

The DMP analysis does not seem that relevant, as there is very little evidence that single methylation polymorphisms are biologically meaningful, and thus could be deleted from the manuscript.

We completely agree that the DMP analysis in almost all methylation studies has shown that single-‐site polymorphisms in plants are of limited biological relevance. Here, we wanted to make these points clear to show that the stress-‐related signal was much stronger for DMRs than for DMPs. We hereby also illustrate why a previous study, analyzing a similar dataset of single cytosines, came to the conclusion that salt stress altered methylation at CG sites (Jiang et al. 2014). As suggested by reviewers, we have reduced this section to a single paragraph and a single supplementary figure. Therefore, we would like to keep these data as they add significant value to the manuscript.

*8) Correlations are made between changes in DNA methylation and H3K27 methylation seen in a previously published study. How comparable are the stress treatments between that study and this, in addition to the tissues profiled? It is not clear what mechanistic relationship is being proposed. Is it possible to test P1/P2 transgenerational stress responses in mutants in the H3K27 pathway? This section didn't really add much to an already complex and data-rich paper.*

It is possible to test the transgenerational stress responses in mutants affecting the H3K27 pathway, although the signal can be bit weaker with some mutants in this pathway having severe growth phenotypes. Nevertheless, we believe the association between methylation and H3K27me3 deposition at HS-DMRs will be an exciting area of future investigation.

9) Throughout the text it needs to be made clearer what experiments the authors performed and what experiments are from published data and were not performed under the same growth regime. Examples of this include the chromatin profiling from hyperosmotic stressed plants (Table 1), and transcriptional profiling upon hyperosmotic treatment (Figure 5—figure supplement 1). Furthermore, the reasons why the salt treatments vary for different experiments should be explained – i.e. 25 or 75 mM for the initial experiments, for 6 weeks, but then 200 mM for the germination test but 150 mM for the survival test (for 14 days each), Chlorophyll content used 100 mM for 5 weeks, and the RNA analysis used 125 mM for 5 weeks. The authors might also mention which, if any, of these levels are considered reasonable for salt stress in nature.

We have included references to all the public datasets used. The levels of salt employed for the different experiments have been explained in the Methods section. In nature, the concentration of salt in soil could range from 5-‐600 nM but we decided to use treatments (25-‐75 mM) already shown to be efficient in inducing adaptive responses in *Arabidopsis* (Sani et al., 2014).

10) Certain classes of transposons are identified as being enriched in DMRs associated with salt treatment. The locus-specific blow ups in Figure 5A were convincing, but the authors should consider providing some other specific loci and transposons that serve as representatives of the changes observed, alongside the summary graphs.

Our detailed data analyses will be made publicly available through a web browser (EPIC-CoGe, supported by Araport and iPlant; http://genomevolution.org/r/939v). Thus, readers will be able to look up genomic loci of their choice and we avoid including additional supplemental data to an already data-rich manuscript.

11) In Figure 2B, gain of non-CG methylation appears to be the strongest effect. Is the model that this is stress triggered de novo DNA methylation? The authors should consider repeating part of this experiment in a drm1drm2 mutant, which should abolish these changes if they are mediated via the canonical RdDM pathway. Demonstrating the genetic dependence of this effect on de novo methylation would strengthen the argument.

We have performed this experiment. Salt-‐induced DMRs (in wild-type) do not show hyper-methylation in *drm1 drm2* mutants, supporting the notion that hyper-‐methylation depends on the RdDM pathway. Results have been included in new Figure 3—figure supplement 2.